# Signaling pathways as linear transmitters

**Harry Nunns\*, Lea Goentoro\***

Division of Biology and Biological Engineering, California Institute of Technology, Pasadena, United States

**Abstract** One challenge in biology is to make sense of the complexity of biological networks. A good system to approach this is signaling pathways, whose well-characterized molecular details allow us to relate the internal processes of each pathway to their input-output behavior. In this study, we analyzed mathematical models of three metazoan signaling pathways: the canonical Wnt, MAPK/ERK, and Tgfβ pathways. We find an unexpected convergence: the three pathways behave in some physiological contexts as linear signal transmitters. Testing the results experimentally, we present direct measurements of linear input-output behavior in the Wnt and ERK pathways. Analytics from each model further reveal that linearity arises through different means in each pathway, which we tested experimentally in the Wnt and ERK pathways. Linearity is a desired property in engineering where it facilitates fidelity and superposition in signal transmission. Our findings illustrate how cells tune different complex networks to converge on the same behavior.

DOI: https://doi.org/10.7554/eLife.33617.001

## Introduction

Cells must continually sense, interpret, and respond to their environment. This is orchestrated by signaling pathways: networks of multiple proteins that transmit signals and initiate cellular response. Signaling pathways are critical to animal development and physiology, and yet there are fewer than 20 classes of metazoan signaling pathways (*Gerhart, 1999*). These signaling pathways evolved prior to the Cambrian and remain highly conserved across animal phyla (*Gerhart, 1999*; *Pires-daSilva and Sommer, 2003*). Each signaling pathway, therefore, governs a wide range of cellular events, both within and across organisms.

Insights into the versatility of signaling pathways may be gleaned from pathway architectures. Indeed, distinct architectural features define each pathway. Studies over the past several decades have revealed distinct signaling capabilities that arise from pathway architecture, for example, all-or-none response in the MAPK/ERK pathway (*Huang and Ferrell, 1996*; *Ferrell and Machleder, 1998*), oscillations in the NFκB pathway (*Hoffmann et al., 2002*), or asymmetrical cell signaling in the Notch/Delta pathway (*Sprinzak et al., 2010*). Alternatively, analysis of pathway architectures may also reveal shared signaling capabilities that emerge from the distinct architectures, pointing to a fundamental property that pathways have converged upon despite their separate evolutionary trajectories. In this study, we sought to identify shared properties between conserved signaling pathways.

To this end, we examined three signaling pathways, the canonical Wnt, ERK and Tgfβ pathways. These pathways are activated by an extracellular ligand binding to a membrane receptor (*Figure 1A*). The ligand-receptor activation initiates a series of biochemical reactions within the cell, culminating in a buildup of transcriptional regulator, which regulates transcription of broad gene targets. Since the ligand-receptor module is relatively plastic across organisms (e.g. flies have one EGF receptor whereas humans have four [*Citri et al., 2003*]), we focused on the conserved core pathway (*Figure 1A*). We define the input to the core pathway as the ligand-receptor activation, and the output as the level of transcriptional regulator.

**\*For correspondence:**
hnunns@caltech.edu (HN);
goentoro@caltech.edu (LG)

**Competing interests:** The authors declare that no competing interests exist.

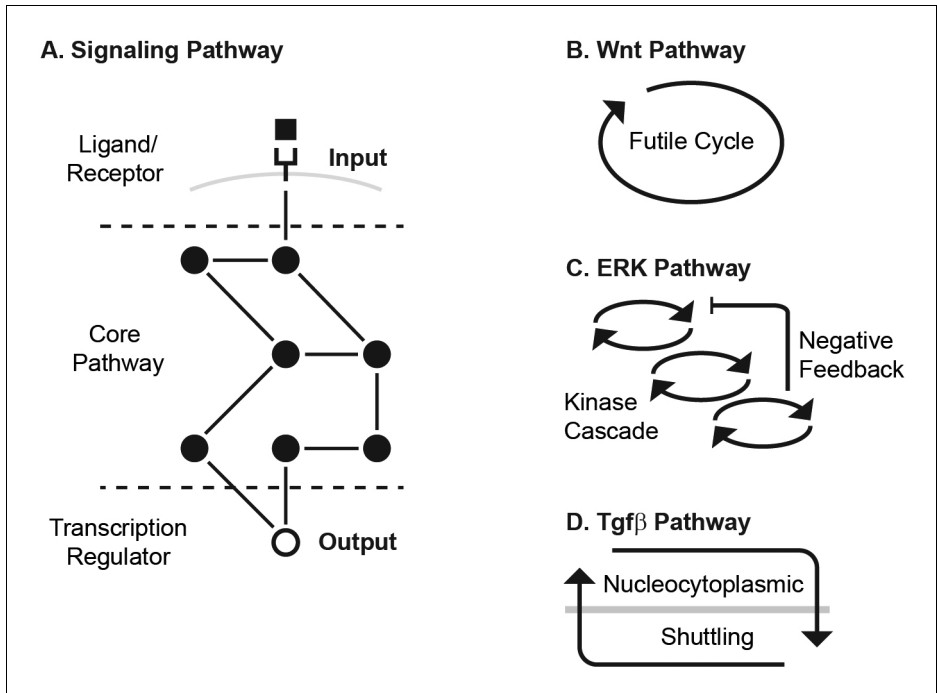

**Figure 1.** The Wnt, ERK, and Tgfβ pathways transmit input using different core transmission architecture. (A) Signaling pathways transmit inputs from ligand-receptor interaction to a change in output, the level of transcriptional regulator (white circle). (B-D) The core pathway for each metazoan signaling pathway is defined by distinct architectural features. In the Wnt pathway (B), the output is regulated by a futile cycle of continual synthesis and rapid degradation. In the ERK pathway (C), the output is regulated by a kinase cascade coupled to negative feedback. In the Tgfβ pathway (D), the output is regulated through continual nucleocytoplasmic shuttling.
DOI: https://doi.org/10.7554/eLife.33617.002

The Wnt, ERK, and Tgfβ pathways transmit input using different core transmission architecture (*Figure 1B–D*). In the Wnt pathway, signal transmission is characterized by a futile cycle of synthesis and rapid degradation (*Kimelman and Xu, 2006*; *Saito-Diaz et al., 2013*; *Hoppler and Moon, 2014*). We use the term futile cycle to highlight that β-catenin is continually synthesized only to be quickly targeted for degradation and kept at low concentration, as opposed to, for instance, being synthesized only as needed. Ligand-receptor input diminishes the degradation arm of this cycle, leading to accumulation of β-catenin output (*Kimelman and Xu, 2006*; *Stamos and Weis, 2013*; *Nusse and Clevers, 2017*). In the ERK pathway, signal transmission is characterized by a cascade of phosphorylation events coupled to feedbacks, leading to an increase in phosphorylated ERK output (*Kolch, 2005*; *Yoon and Seger, 2006*; *Avraham and Yarden, 2011*; *Lake et al., 2016*). Finally, signal transmission in the Tgfβ pathway is characterized by continual nucleocytoplasmic protein shuttling (*Inman et al., 2002*; *Nicolás et al., 2004*; *Xu and Massagué, 2004*; *Schmierer and Hill, 2005*; *Massagué et al., 2005*). Ligand-receptor input effectively increases the rate of nuclear import, leading to an increase in output, the nuclear Smad complex (*Schmierer et al., 2008*).

Importantly for our approach, the architectures of the three pathways are captured by mathematical models that have been refined by years of experiments. Although by no means complete, the mathematical models have track records of success in predicting systems-level behaviors across multiple biological systems. For instance, the Wnt model (*Lee et al., 2003*) captures the dynamics of destruction complex well enough as to enable prediction of robustness in fold-change response (*Goentoro and Kirschner, 2009*) and the differential roles of the two scaffolds in the pathway (*Lee et al., 2003*); the ERK model (*Huang and Ferrell, 1996*; *Ferrell and Bhatt, 1997*; *Schoeberl et al., 2002*; *Sturm et al., 2010*) captures the ultrasensitivity in the phosphorylation cascade (*Huang and Ferrell, 1996*); and the Tgfβ model (*Schmierer et al., 2008*) reveals the roles of

nucleocytoplasmic shuttling in transducing the duration and intensity of ligand stimulation (*Schmierer et al., 2008*).

We studied these mathematical models to identify what, if any, behaviors converge across pathways. The Wnt (*Lee et al., 2003*), ERK (*Sturm et al., 2010*), and Tgfβ (*Schmierer et al., 2008*) models consist of 7, 26, and 10 coupled, nonlinear ODEs, respectively, with 22, 46, and 13 parameters. Because of their large sizes, they are typically solved numerically to simulate experimental observations and generate new predictions. However, for the questions posed here, we found that numerical simulations are not sufficient. Rather, we needed analytics to uncover exactly how the pathway behaviors depend on the underlying biochemical processes. While we previously derived an analytical solution to the Wnt pathway (*Goentoro and Kirschner, 2009*), analytical treatment of the Tgfβ and ERK pathways has not been attempted due to the complex, nonlinear equations involved. To address this problem, we employed various analytical techniques, including graph theory-based variable elimination and dimensional analysis, to derive analytical or semi-analytical solutions to the steady-state output of each pathway. Our analysis, along with subsequent experimental verification, reveals a striking convergence across the Wnt, Tgfβ, and ERK pathways: cells operate in the parameter regime where the complex, nonlinear interactions in each pathway give rise to linear signal transmission.

## Results

### Mathematical analysis identifies the Wnt, ERK, and Tgfβ pathway as linear transmitters

We began our analysis using established models of the Wnt (*Lee et al., 2003*), ERK (*Sturm et al., 2010*), and Tgfβ (*Schmierer et al., 2008*) pathways. These models capture the salient features of each pathway, and include biochemical details such as synthesis, degradation, binding, dissociation and post-translational modifications. In all the models, biochemical parameters have been directly measured or fitted to kinetic measurements from cell, embryo or extract systems. Numerical simulation of each model has predicted a wide range of pathway behaviors over the years (e.g. Wnt refs. [*Lee et al., 2003*; *Goentoro and Kirschner, 2009*; *Hernández et al., 2012*]; ERK refs. [*Huang and Ferrell, 1996*; *Ferrell and Machleder, 1998*; *Schoeberl et al., 2002*; *Sturm et al., 2010*; *Fritsche-Guenther et al., 2011*]; Tgfβ refs. [*Schmierer et al., 2008*; *González-Pérez et al., 2011*; *Andrieux et al., 2012*; *Vizán et al., 2013*; *Wang et al., 2014*]). Below, we describe our analysis of each pathway and the unifying behavior that emerges from all three pathways.

### Canonical Wnt pathway

In this pathway, cells sense ligand-receptor input by monitoring β-catenin protein (*Kimelman and Xu, 2006*; *Stamos and Weis, 2013*; *Nusse and Clevers, 2017*; *MacDonald et al., 2009*; *Clevers and Nusse, 2012*). β-catenin is continually synthesized and rapidly degraded by a large destruction complex, comprised of multiple proteins including APC, Axin, and GSK3β. The destruction complex binds and phosphorylates β-catenin, tagging it for degradation by the ubiquitin/proteosome machinery (*Kimelman and Xu, 2006*; *Stamos and Weis, 2013*). Wnt ligands, through binding to Frizzled and LRP receptors, inhibit the destruction complex, leading to accumulation of β-catenin. β-catenin then regulates the expression of broad target genes (*Stamos and Weis, 2013*; *Nusse and Clevers, 2017*).

The model of the Wnt pathway (*Figure 2A*) was published in 2003 by a collaboration between the Kirschner and Heinrich labs (*Lee et al., 2003*). The Wnt model consists of seven nonlinear differential equations and 22 parameters. Applying dimensional analysis, we previously derived the analytical solution to β-catenin concentration at steady-state (*Goentoro and Kirschner, 2009*):

$$[\beta\text{cat}]_{\text{ss}} = K_{17} \cdot \frac{1 - \gamma + \frac{\alpha}{u}}{2}\left(\sqrt{1 + \frac{4\gamma}{\left(1 - \gamma + \frac{\alpha^2}{u}\right)}} - 1\right) \tag{1}$$

$$\alpha = \frac{k_4 k_6 k_9 v_{14} \cdot \text{GSK3}_{\text{tot}} \cdot \text{APC}_{\text{tot}}}{k_5 k_{-6} K_7 K_8 k_{13} k_{15}} \tag{2}$$

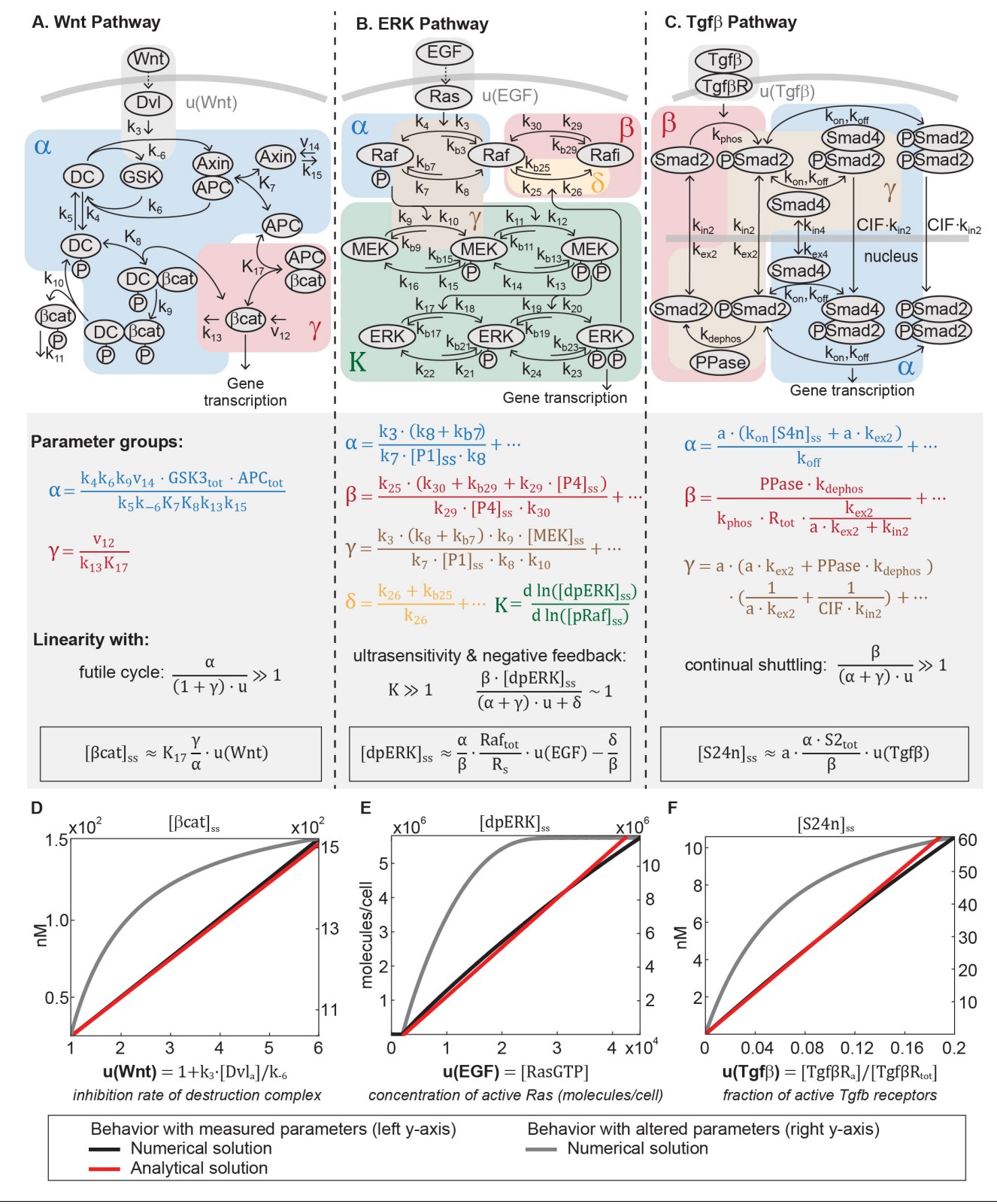

**Figure 2.** The Wnt, ERK, and Tgfβ pathways are linear signal transmitters. (A-C) Network diagrams of the signaling pathways. The Tgfβ diagram is modified from *Schmierer et al. (2008)*. In the network diagram in A, DC refers to the β-catenin destruction complex. Below the network diagrams: the parameter groups and linearity equations we analytically derived in this study. Parameter groups and input functions are color-coded to the corresponding reactions in the network diagrams. Parameters that do not appear in the parameter groups either drop out due to irreversible reaction

*Figure 2 continued on next page*

*Figure 2 continued*

steps (such as $k_{10}$ and $k_{11}$ in the Wnt pathway) or negligible (as indicated by ellipses). (**D-F**) Our analysis reveals that in physiologically relevant parameter values, these pathways generate a linear input-output relationship. The outputs are β-catenin, dpERK, and nuclear Smad complex for the Wnt, ERK, and Tgfβ pathway, respectively. The input functions u describe the effect of ligand-receptor interactions on the core pathway. Specifically: u(Wnt) is the rate by which Dishevelled/Dvl inhibits the destruction complex upon Wnt ligand activation, where $k_3$ and $k_{-6}$ are defined in the figure and $[Dvl]_a$ is the concentration Wnt-activated Dishevelled (see *Equations A15*); u(EGF) is concentration of EGF-activated Ras (Ras-GTP); and u(Tgfβ) is the fraction of Tgfβ -activated receptors. Red and blue lines, respectively: analytical and numerical solutions with measured parameters (plotted against the left y-axis). Grey line: examples of numerical solutions outside measured parameters (plotted against the right y-axis).

DOI: https://doi.org/10.7554/eLife.33617.003

The following source data and figure supplements are available for figure 2:

**Source code 1.**

DOI: https://doi.org/10.7554/eLife.33617.011

**Figure supplement 1.** Model simulations for the ERK pathway.

DOI: https://doi.org/10.7554/eLife.33617.004

**Figure supplement 2.** The predicted linearity extends throughout the dynamic range of the ERK and Tgfβ pathways.

DOI: https://doi.org/10.7554/eLife.33617.005

**Figure supplement 3.** Model simulations for the Tgfβ pathway.

DOI: https://doi.org/10.7554/eLife.33617.006

**Figure supplement 4.** Incorporating into the Wnt model the dual function of GSK3β in phosphorylating β-catenin and LRP5/6.

DOI: https://doi.org/10.7554/eLife.33617.007

**Figure supplement 5.** The requirements for linear signal transmission in the Wnt, Tgfβ, and ERK pathway.

DOI: https://doi.org/10.7554/eLife.33617.008

**Figure supplement 6.** Linear signal transmission occurs over a range of parameters in the model.

DOI: https://doi.org/10.7554/eLife.33617.009

**Figure supplement 7.** Numerical simulation of the input-output relationship of the NF-κB pathway.

DOI: https://doi.org/10.7554/eLife.33617.010

$$\gamma = \frac{v_{12}}{k_{13} K_{17}} \qquad (3)$$

where the input function $u = u(Wnt)$ is the rate of inhibition of the destruction complex (DC) via Dishevelled/Dvl, a function of ligand-receptor activation. As illustrated in *Figure 2A*, $K_i$'s are equilibrium dissociation constants, $k_i$'s are rate constants, and $v_i$'s are synthesis rates. $\alpha$ and $\gamma$ in *Equation 1* are dimensionless parameter groups defined in *Equations 2 and 3*: $\alpha$ characterizes β-catenin degradation by the destruction complex, and $\gamma$ characterizes the extent to which β-catenin binds to APC independently of the destruction complex.

*Equation 1* demonstrates that, in general, β-catenin concentration is a nonlinear function of the input u. Many parameters of the model were directly measured in *Xenopus* extracts, and the remaining calculated from measurements in the same system (*Appendix 1—table 1*). In this study, we examined how the analytical solution (*Equation 1*) behaves with these measured parameters. The measured parameters (*Appendix 1—table 1*) indicate that $\alpha \sim 66$, $\gamma \sim 1.4$, and for maximal stimulation, $u \sim 6.0$. The large $\alpha$ reflects how β-catenin stability is primarily dictated by the destruction complex, that is, $\alpha/u \gg 1$ means that non-Axin-dependent degradation is minimal, and $\alpha/u \gg \gamma$ means that the positive feedback from sequestration by APC is minimal. Indeed, the rapid action of the destruction complex in the Wnt pathway is a recurring observation across biological systems (*Kimelman and Xu, 2006*; *Saito-Diaz et al., 2013*; *Hoppler and Moon, 2014*). With $\alpha/u \gg 1 + \gamma$, *Equation 1* simplifies to

$$[\beta cat]_{ss} \approx K_{17} \frac{\gamma}{\alpha} u \qquad (4)$$

with detailed derivations presented in Appendix 1. Therefore, within physiologically relevant parameter values, the steady-state β-catenin concentration becomes a linear function of the input u (red line, *Figure 2D*). The linear input-output relationship holds for the entire dynamic range of the model, until the system saturates at maximal stimulation ($u \sim 6.0$). We confirmed that the numerical solution of the full model matches the analytical solution in *Equation 4* (blue line, *Figure 2D*), and

that the response becomes nonlinear when $\alpha$ is decreased, breaking the requirement $\alpha/u \gg 1 + \gamma$ (grey line, *Figure 2D*).

Source codes for the numerical simulations in *Figure 2D–F* (grey and black lines) are available in *Figure 2—source code 1*.

## ERK pathway

The unexpected linearity that emerges from the model of the Wnt pathway prompted us to wonder if such simplicity may be found in other pathways. Strikingly, we observed the same linearity in the ERK and Tgfβ pathways. In the ERK pathway (*Figure 2B*), ligand-receptor input is transmitted via a cascade of protein phosphorylation (*Kolch, 2005*; *Yoon and Seger, 2006*). In particular, ligand-receptor interactions activate Ras, which leads to membrane recruitment and phosphorylation of Raf. Phosphorylated Raf subsequently doubly phosphorylates MEK, which in turn doubly phosphory-lates ERK (*Kolch, 2005*). Doubly-phosphorylated ERK (dpERK) is a transcriptional regulator that affects a broad array of genes (*Yoon and Seger, 2006*). The multi-step topology of the kinase cas-cade, combined with distributive phosphorylation of each kinase, gives rise to ultrasensitivity – first demonstrated in the seminal work by the Ferrell lab (*Huang and Ferrell, 1996*; *Ferrell and Machleder, 1998*). In other contexts, the pathway also exhibits a graded response (*Whitehurst et al., 2004*; *Mackeigan et al., 2005*; *Cohen-Saidon et al., 2009*; *Ahmed et al., 2014*) that is thought to arise from the incorporation of negative feedbacks (*Lake et al., 2016*), one of which is the inhibition of Raf by dpERK through hyper-phosphorylation of serine residues (*Sturm et al., 2010*; *Dougherty et al., 2005*; *Hekman et al., 2005*).

The ERK model (*Sturm et al., 2010*) is the product of more than two decades of refinement (*Huang and Ferrell, 1996*; *Ferrell and Machleder, 1998*; *Schoeberl et al., 2002*; *Sturm et al., 2010*; *Fritsche-Guenther et al., 2011*). The model, which captures ultrasensitivity and Raf feedback, consists of 26 differential equations and 46 parameters. To derive an analytical expression for the ERK pathway, we used a variable elimination technique developed for networks of mass action kinet-ics (*Feliu and Wiuf, 2012*). The technique utilizes an algebraic framework, linear elimination of varia-bles, and mass conservation laws to parameterize steady-state in terms of core variables (described in Appendix 1). We derived an analytical relationship between the steady-state output of the path-way $[\text{dpERK}]_{ss}$ and the input to the phosphorylation cascade $u$:

$$[\text{dpERK}]_{ss} = \frac{\alpha}{\beta} \cdot \left( \frac{\text{Raf}_{\text{tot}}}{[\text{pRaf}]_{ss}} \right) - 1 - \frac{\gamma}{\alpha} \cdot u - \frac{\delta}{\beta} \tag{5}$$

$$\alpha = \frac{k_3 \cdot (k_8 + k_{b7})}{k_7 \cdot [P1]_{ss} \cdot k_8} + \cdots \tag{6}$$

$$\beta = \frac{k_{25} \cdot (k_{30} + k_{b29} + k_{29} \cdot [P4]_{ss})}{k_{29} \cdot [P4]_{ss} \cdot k_{30}} + \cdots \tag{7}$$

$$\gamma = \frac{k_3 \cdot (k_8 + k_{b7}) \cdot k_9 \cdot [\text{MEK}]_{ss}}{k_7 \cdot [P1]_{ss} \cdot k_8 \cdot k_{10}} + \cdots \tag{8}$$

$$\delta = \frac{k_{26} + k_{b25}}{k_{26}} + \cdots \tag{9}$$

Detailed derivations of *Equation 5* are presented in Appendix 1. The input $u = u(\text{EGF})$ in *Equa-tion 5* is the concentration of active Ras, which is activated via GTP loading at the ligand-receptor complex (*Kolch, 2005*). The parameter groups $\alpha$, $\beta$, $\gamma$, and $\delta$ in *Equation 5* are defined in *Equa-tions 6–9*, where the ellipses indicate additional small terms (expanded in Appendix 1). The relative magnitudes of $\alpha$, $\beta$, $\gamma$, and $\delta$ indicate how the Raf pool partitions during signaling (*Equations A21*, *A29–A31*). The dimensionless group $\alpha \cdot u$ relates to the amount of free, phosphorylated Raf ($\alpha$, blue-shaded in *Figure 2B*), $\beta \cdot [\text{dpERK}]_{ss}$ describes the amount of Raf inhibited through negative feedback by dpERK ($\beta$, red-shaded in *Figure 2B*), $\delta$ relates to the amount of unphosphorylated ($\delta$, blue-shaded in *Figure 2B*), and $\gamma \cdot u$ relates to the amount of phosphorylated Raf bound to other proteins

(e.g. to MEK, brown-shaded in *Figure 2B*). *Equation 5* is not a closed solution, as it includes the term $[\mathrm{pRaf}]_{\mathrm{ss}}$, and there are variables included in parameter groups $\alpha$, $\beta$, $\gamma$. We confirmed that the parameter groups remain constant over the course of signaling (within 10%, *Figure 2—figure supplement 1*), justifying treating the latter variables as parameters.

Next, we considered how the analytical expression (*Equation 5*) behaves within a specific parameter regime observed in experiments. First, experiments in several mammalian cell systems have shown that feedback is strong, such that a significant fraction of the Raf pool is inhibited (*Fritsche-Guenther et al., 2011*; *Dougherty et al., 2005*). This means that $\beta \cdot [\mathrm{dpERK}]_{\mathrm{ss}} \sim (\alpha + \gamma) \cdot \mathrm{u} + \delta$. Second, as has been observed in multiple contexts ([*Huang and Ferrell, 1996*; *Ferrell and Machleder, 1998*; *Schoeberl et al., 2002*; *Sturm et al., 2010*] *Appendix 1—table 2*), ERK phosphorylation is ultrasensitive to the amount of pRaf (the ultrasensitive cascade is shaded green in *Figure 2B*). Denoting $\mathrm{K}$ as the relative change of $[\mathrm{dpERK}]_{\mathrm{ss}}$ with respect to $[\mathrm{pRaf}]_{\mathrm{ss}}$, ultrasensitivity entails that $\mathrm{K} \gg 1$. In this range, small changes in pRaf level have very large effects on dpERK level (e.g., in model simulations, a 30% change in pRaf level results in a 900% change in dpERK level, *Figure 2—figure supplement 1*). We find analytically that in the parameter regime where $\beta \cdot [\mathrm{dpERK}]_{\mathrm{ss}} \sim (\alpha + \gamma) \cdot \mathrm{u} + \delta$ and $\mathrm{K} \gg 1$, the negative feedback holds the level of pRaf constant ($[\mathrm{pRaf}]_{\mathrm{ss}} \approx \mathrm{R_s}$, details in Appendix 1). With these two features, strong negative feedback and ultrasensitivity, dpERK becomes a linear function of the input $\mathrm{u}$:

$$[\mathrm{dpERK}]_{\mathrm{ss}} \approx \frac{\alpha}{\beta} \cdot \frac{\mathrm{Raf_{tot}}}{\mathrm{R_s}} \cdot \mathrm{u} - \frac{\delta}{\beta} \tag{10}$$

The full derivation is given in Appendix 1, and includes a toy model to illustrate the intuition for how ultrasensitivity combines with negative feedback to produce linearity. *Equation 10* is plotted in *Figure 2E* (red line). We confirmed that the numerical solution of the full model matches the analytics in *Equation 10*, and becomes nonlinear when the negative feedback is weakened (grey line, *Figure 2E*). Although the analytical expression describes up until 50% of ERK activation, we verified numerically that the predicted linearity extends to 93% of ERK activation (*Figure 2—figure supplement 2*).

The linearity derived here applies across different dynamic ERK responses. The model we analyzed gives a sustained dpERK response. In some contexts, however, the ERK pathway shows a pulsatile response, which has been attributed to receptor desensitization (*Schoeberl et al., 2002*). Using a larger model that includes details of receptor desensitization (*Schoeberl et al., 2002*), we numerically verified that the linearity holds for pulsatile responses - that is, the peak level of dpERK increases linearly with the peak level of $\mathrm{u}$ (*Figure 2—figure supplement 1*).

## Tgfβ pathway

Finally, we examined signal transduction within the Tgfβ pathway (*Figure 2C*). In the Tgfβ pathway, input from ligand-receptor interactions is transmitted by the Smad proteins. There are several classes of Smad proteins, including the receptor-regulated Smads (R-Smads) and the common Smad (co-Smad or Smad4) (*Massagué et al., 2005*). Ligand-activated receptors phosphorylate R-Smads. Phosphorylated R-Smads bind to the co-Smad, and shuttle into the nucleus and regulate broad target genes. In the nucleus, the Smad complex dissociates and R-Smads are constitutively de-phosphorylated and shuttled out to the cytoplasm, where the cycle of phosphorylation and complex formation begins again (*Schmierer et al., 2008*). This dynamic translocation in and out of the nucleus forms a continual nucleocytoplasmic shuttling of Smads, a known integral feature of the Tgfβ pathway (*Inman et al., 2002*; *Nicolás et al., 2004*; *Xu and Massagué, 2004*; *Schmierer and Hill, 2005*).

The Tgfβ model (*Schmierer et al., 2008*) was published in 2008 by the Hill lab, and consists of 10 differential equations and 13 parameters. Even though the model was fitted to R-Smad2 data, the general architecture of signal transmission is conserved across all five R-Smads (*Massagué et al., 2005*; *Schmierer et al., 2008*). Using the variable elimination technique described before (*Feliu and Wiuf, 2012*), we derived an analytical expression of the steady-state concentration of Smad complex in the nucleus:

$$[\mathrm{S24n}]_{\mathrm{ss}} = \mathrm{a} \cdot \frac{\alpha \cdot \mathrm{u}}{(\alpha + \gamma) \cdot \mathrm{u} + \beta} \mathrm{S2_{tot}} \tag{11}$$

$$\alpha = \frac{a \cdot (k_{on}[S4n]_{ss} + a \cdot k_{ex2})}{k_{off}} + \cdots \tag{12}$$

$$\beta = \frac{PPase \cdot k_{dephos}}{k_{phos} \cdot R_{tot} \cdot \dfrac{k_{ex2}}{a \cdot k_{ex2} + k_{in2}}} + \cdots \tag{13}$$

$$\gamma = a \cdot (a \cdot k_{ex2} + PPase \cdot k_{dephos}) \left( \frac{1}{a \cdot k_{ex2}} + \frac{1}{CIF \cdot k_{in2}} \right) + \cdots \tag{14}$$

In *Equation 11*, the input function $u = u(Tgf\beta)$ is the active fraction of Tgfβ receptors. The parameter $a$ is the nucleocytoplasmic volume ratio. The dimensionless parameter groups $\alpha$, $\beta$, and $\gamma$ in *Equation 11* are defined in *Equations 12–14*, where the ellipses indicate additional small terms (expanded in Appendix 1). $\alpha$, $\beta$, and $\gamma$ describe how the Smad2 pool partitions during signaling (*Equations A44, A50, A51*): $\alpha \cdot u$ relates to the amount of nuclear Smad complex ($\alpha$, blue-shaded in *Figure 2C*, captures the parameters related to complex formation and translocation to the nucleus), $\beta$ relates to the amount of free, unphosphorylated Smad2 ($\beta$, red-shaded in *Figure 2C*, captures the parameters related to complex dissociation and translocation to the cytoplasm), and $\gamma \cdot u$ loosely relates to the remaining Smad2 pool ($\gamma$ is brown-shaded in *Figure 2C*). Phosphorylated Smad2 quickly forms complex (*Lagna et al., 1996*), so $\beta$ essentially corresponds to total monomeric Smad2. Finally, *Equation 11* is not a closed solution, since variable $[S4n]_{ss}$ appears in $\alpha$. We numerically tested that it is constant within 2% for non-saturating inputs (*Figure 2—figure supplement 3*), justifying treating it as a parameter.

As in the Wnt and ERK pathway, the analytical expression for nuclear Smad complex (*Equation 11*) allows us to see that the behavior dramatically simplifies with parameters observed in experiment. We consider the case for non-saturating inputs ($u \sim 0.1$). Protein concentrations in the Tgfβ model were measured in human keratinocyte cells and the rate constants fitted to kinetic data measured in the cells (*Schmierer et al., 2008*). With the measured parameters (*Appendix 1—table 3*), we find that $\beta \sim 46$, $\alpha \cdot u \sim 1.5$, and $\gamma \cdot u \sim 0.7$. In this parameter regime, once Smad2 is imported to the nucleus, it is rapidly dephosphorylated and exported. Dynamic Smad2 translocation maintains monomeric Smad2 in excess to Smad complex ($\beta \gg (\alpha + \gamma) \cdot u$). and forms the continual nucleocytoplasmic shuttling that is characteristic of the Tgfβ pathway. Even under maximal Tgfβ stimulation, it has been estimated that phosphorylated Smad2 comprises only 36% of the Smad2 pool (*Schmierer and Hill, 2005*; *Gao et al., 2009*). With $\beta \gg (\alpha + \gamma) \cdot u$, the first term in the denominator of *Equation 11* is small, and concentration of nuclear Smad complex becomes a linear function of input:

$$[S24n]_{ss} \approx a \cdot \frac{\alpha \cdot S2_{tot}}{\beta} \cdot u \tag{15}$$

*Equation 15* is plotted in *Figure 2F* (red line), and we confirmed that numerical simulations recapitulates *Equation 15* (blue line, *Figure 2F*). Although the analytical solution is valid only for small values of u, we numerically verified that the predicted linearity holds for the entire range of input u (from 0 to 1, *Figure 2—figure supplement 2*). We confirmed that the pathway becomes nonlinear when the R-Smad phosphatase is inhibited such that $\beta \sim (\alpha + \gamma) \cdot u$ (grey line, *Figure 2F*). While the model analyzed here gives a sustained Smad response, we verified numerically that the linearity holds for a larger model that includes receptor desensitization and gives a pulsatile Smad response (*Figure 2—figure supplement 3*) (*Vizán et al., 2013*).

## Linearity in the Wnt and ERK pathways was observed experimentally

Analytical expressions for the Wnt, ERK, and Tgfβ pathways reveal that the three pathways behave as linear signal transmitters within parameter regimes measured in cells. To confirm the linearity, we directly measured the input-output relationships in human cell lines. We focused our efforts on the Wnt and ERK pathways, since we are limited by available antibodies in the Tgfβ pathway.

To analyze the canonical Wnt pathway, we performed quantitative Western blot measurements in RKO cells, a model system for Wnt signaling. To track the input, we measured the level of

phosphorylated LRP5/6 receptors (on Ser1490), which increases within minutes of ligand-receptor complex formation (*Tamai et al., 2004*). To track the output, we measured the level of β-catenin. We confirmed that the level of phosphorylated LRP5/6 and β-catenin increase upon Wnt simulation and reach steady-state within 6 hr (*Figure 3—figure supplement 1*). Accordingly, all subsequent measurements were done at 6 hr after Wnt stimulation.

To measure the input-output relationship in the Wnt pathway, we treated RKO cells with varying doses of purified Wnt3A and measured how β-catenin (output) correlates with phosphorylated LRP (input). As shown in *Figure 3A*, the level of β-catenin increases linearly with the level of phosphorylated LRP. The linearity persists until saturation of the input, defined as 90% of maximal phosphorylated LRP response (blue circles, *Figure 3A*; *Figure 3—figure supplement 2*). Notably, at high doses of Wnt3A, β-catenin continues to show incremental activation, despite saturation in phosphorylation of LRP (grey circles, *Figure 3A*). This can be explained within some findings that, while Frizzled/LRP complex is the primary receptor input in β-catenin activation, β-catenin can be activated independently of LRP (e.g. *Rotherham and El Haj, 2015*).

Consistent with the mathematical analysis, we observed in RKO cells that the Wnt pathway behaves as a linear transmitter throughout the dynamic range of the input. As a control that is expected from the Michaelis-Menten kinetics that describe ligand binding in the model, we confirmed that the linearity does not extend upstream to Wnt dose: both phospho-LRP5/6 and β-catenin show nonlinear response to Wnt dose (*Figure 3—figure supplement 2*). Therefore, in the Wnt pathway, a nonlinear ligand-receptor processing step is followed by linear signal transmission through the core intracellular pathway.

Next, to measure the input-output relationship in the ERK pathway, we performed quantitative Western blots in H1299 cells, one of the model systems used in the field. Linearity in the ERK pathway has been suggested in different parts of the pathway, e.g. *Knauer et al. (1984)* used experimental and modeling analyses to infer linearity between receptor occupancy and the downstream cellular proliferation; *Oyarzún et al. (2014)* suggests linearity in ligand-receptor processing. Here, we specifically probe linearity in the core transmission step of the pathway. Detecting the input level, EGF-activated Ras GTP, requires a pull down step that makes it less quantifiable. Therefore, motivated by *Oyarzún et al. (2014)*, we tested EGF ligand itself as the input. To track the output, we measured the level of doubly-phosphorylated ERK1/2 (on Thr202/Tyr204), dpERK. We first characterized the kinetics of response: dpERK peaks 5 min after EGF stimulation (*Figure 3—figure supplement 3*), and saturates at 4 ng/ml EGF (grey circles, *Figure 3B*). Accordingly, all subsequent measurements were performed at 5 min after EGF stimulation, and linearity was assessed over the input range of 0–4 ng/mL EGF (blue circles, *Figure 3B*).

We observed linearity in the input-output relationship of the ERK pathway, with the level of dpERK increasing linearly with EGF dose (*Figure 3B*). The linearity holds throughout the dynamic range of the system, over at least 12-fold activation of dpERK. As the ERK pathway is sometime observed to show bimodal response that would be masked by bulk measurements, we confirmed that the H1299 cells indeed show to graded dpERK response in single-cell level (*Figure 3—figure supplement 4*), in agreement with a previous single-cell, live imaging study (*Cohen-Saidon et al., 2009*). Therefore, as in the Wnt pathway, signals are transmitted linearly in the ERK pathway throughout the dynamic range of the cell. Moreover, the linearity in the ERK pathway is more extensive than in the Wnt pathway, as linearity extends all the way upstream, such that the level of dpERK directly reflects the dose of extracellular EGF ligand.

## Linearity in the Wnt and ERK pathways is modulated by perturbation to parameters

Finally, the analytical expressions we derived in this study not only reveal linear signal transmission, but also the mechanisms by which it arises. In the model of the Wnt pathway, linear transmission occurs due to the futile cycle of β-catenin, in the parameter regime where β-catenin is continually synthesized and rapidly degraded (i.e. $\alpha/\mathfrak{u} \gg 1 + \gamma$). This regime is not infinite: for instance, a tenfold decrease in $\alpha$ (e.g. by inhibiting the destruction complex) will break the futile cycle (grey line, *Figure 2D*).

To test if the futile cycle is indeed required for linear signal transmission, we inhibited the destruction complex using CHIR99021, an inhibitor of GSK3β kinase. As before, we measured the input-output relationship, β-catenin vs. phospho-LRP5/6 level, up to 90% of maximal phospho-LRP5/6 input

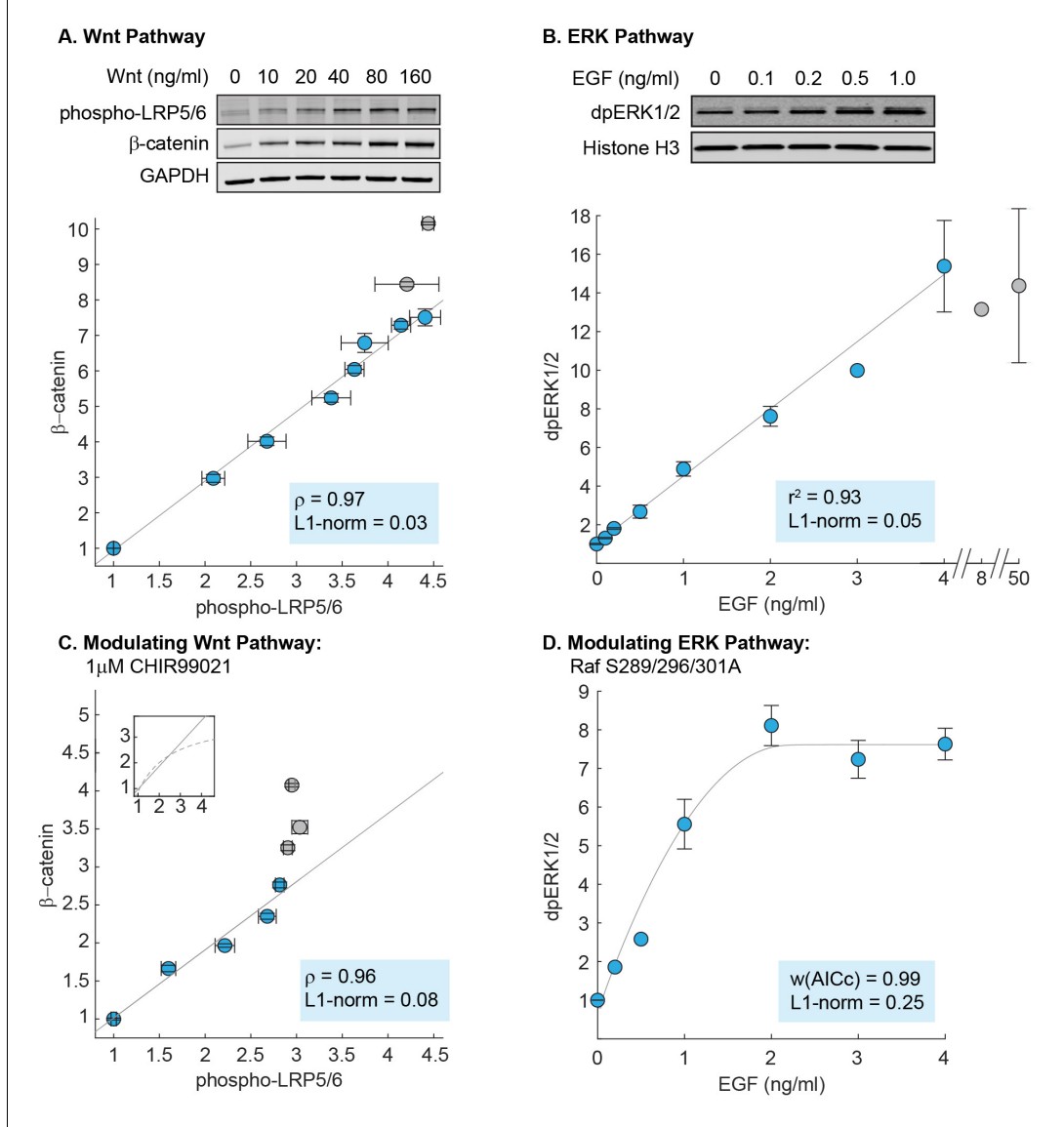

**Figure 3.** Linearity was observed experimentally in the Wnt and ERK pathways. (**A**) Measurements of the input-output relationship in the Wnt pathway. In these experiments, RKO cells were stimulated with 0–1280 ng/mL purified Wnt3A ligand, harvested at 6 hr after ligand stimulation, and lysed for Western blot analyses. Shown on top is a representative Western blot. The data plotted come from seven independent experiments (total N = 66). Each circle indicate the mean intensities of the phospho-LRP5/6 (x-axis) and β-catenin (y-axis) bands for all Western blot biological replicates, and error bars indicate the standard error of the mean. For each gel, we normalize the unstimulated sample (i.e. 0 ng/mL of Wnt3A) to one, and scale the magnitude of the dose response to the average of all gels (described in Materials and methods). The grey line is a least squares regression line, and ρ is the Pearson's coefficient, where ρ = 1 is a perfect positive linear correlation. (**B**) Measurements of the input-output relationship in the ERK pathway. In these experiments, H1299 cells were stimulated with 0–50 ng/mL purified EGF ligand, harvested at 5 min after ligand stimulation, and lysed for Western blot analyses. Shown on top is a representative Western blot. The data plotted here come from five independent experiments (total N = 30). Each circle indicates the mean intensities of dpERK1/2 bands across Western blot biological replicates, and the error bars indicate standard error of the mean. Single replicates are plotted without error bars. All data is plotted relative to unstimulated sample. The grey line is a least squares regression line, and $r^2$ is the coefficient of correlation where $r^2$ = 1 is a perfect linear correlation. (**C**) As in (**A**), except that cells were treated with 1 μM CHIR99021 (detailed in Materials and methods). The data plotted here come from five independent experiments (total N = 59). The grey line is a least squares regression, and ρ is the Pearson's coefficient, where ρ = 1 is a perfect positive linear correlation. Shown in the subplot are the same least squares regression line (solid line), overlaid with the model prediction (dashed line). (**D**) As in (**B**), but measurements were performed in H1299 cells expressing mutant Raf S289/296/301A. The data plotted here come from three independent experiments (total N = 15). The grey line is a fit using the ERK model. We first fitted the gain of the model to the data (i.e. the y-range), and afterward, varied the strength of dpERK feedback ($k_{25}$) to find the best fit. We used the weighted Akaike Information Criterion, w(AICc), to verify that the nonlinear fit from the ERK model outperforms a linear least squares fit (see Materials and methods). 0 < w(AICc) < 1, with higher w(AICc) indicates better performance by the non-linear fit. In all figures, linearity was additionally assessed

*Figure 3 continued on next page*

*Figure 3 continued*

using the least absolute deviations, L1-norm (see Methods). L1-norm can range from 0 to 0.5, with L1-norm < 0.1 indicate a linear relationship. Blue vs grey circles in each figure are explained in the main text. Source files of all Western blot gel images and numerical quantitation data are available in *Figure 3—source data 1*.

DOI: https://doi.org/10.7554/eLife.33617.012

The following source data and figure supplements are available for figure 3:

**Source data 1.**

DOI: https://doi.org/10.7554/eLife.33617.022

**Figure supplement 1.** LRP5/6 phosphorylation and β-catenin accumulation are already at steady state at 6 hr after Wnt stimulation.

DOI: https://doi.org/10.7554/eLife.33617.013

**Figure supplement 2.** The dynamic range of Wnt signaling in RKO cells.

DOI: https://doi.org/10.7554/eLife.33617.014

**Figure supplement 3.** ERK activation peaks at 5 min after EGF stimulation.

DOI: https://doi.org/10.7554/eLife.33617.015

**Figure supplement 4.** Single-cell immunofluorescence measurements show graded ERK response to EGF.

DOI: https://doi.org/10.7554/eLife.33617.016

**Figure supplement 5.** WT Raf-1 overexpression does not affect linear dose-response.

DOI: https://doi.org/10.7554/eLife.33617.017

**Figure supplement 6.** Expression of Raf S29/289/296/301/642A induces non-linear dose-response.

DOI: https://doi.org/10.7554/eLife.33617.018

**Figure supplement 7.** Technical variability from Western blot.

DOI: https://doi.org/10.7554/eLife.33617.019

**Figure supplement 8.** Linearity is not an artifact of loading control normalization.

DOI: https://doi.org/10.7554/eLife.33617.020

**Figure supplement 9.** Linearity was observed across independent experiments.

DOI: https://doi.org/10.7554/eLife.33617.021

(blue circles, *Figure 3C*). As expected, we found that inhibiting the destruction complex (decreasing $\alpha$ in the model) reduced the range of linearity. The non-treated cells (blue circles, *Figure 3A*) exhibit a linear input-output relationship over a 4.4-fold range of LRP input, whereas the CHIR-treated cells show a linear input-output relationship over only a 2.8-fold range of LRP input (blue circles, *Figure 3C*).

Further, our measurements also reveal an unexpected feature of the Wnt pathway. In the model, inhibiting GSK3β causes β-catenin response to become nonlinear for larger inputs (dashed line, *Figure 3C* subplot). In CHIR-treated RKO cells, however, this nonlinearity cannot be reached, as the maximal amount of phosphorylated LRP (input) is reduced by 50% (grey circles, *Figure 3*; *Figure 3— figure supplement 2*), consistent with the dual-function of GSK3β identified by *Zeng et al. (2005)*; *Zeng et al. (2008)* in phosphorylating β-catenin for degradation as well as phosphorylation LRP for activation. Incorporating this dual-role of GSK3β into the model, we found that this expanded model can indeed recapitulate the data (*Figure 2—figure supplement 4*). Therefore, our data indicate two findings: first, that inhibiting GSK3β reduces the range of linear input-output behavior in the Wnt pathway, as predicted by our analytics, and second, that GSK3β co-regulation of β-catenin and LRP unexpectedly constrains the system within the linear regime.

Next, we examine the requirements for linearity in the ERK pathway. *Equation 10* reveals that linearity in the ERK pathway depends upon the coupling of strong nonlinearities – ultrasensitivity and negative feedback. As in the Wnt pathway, this regime is not infinite, for example, decreasing the strength of feedback $\beta$ enables the system to exit the ultrasensitive regime, and therefore reduces linearity (grey line, *Figure 2E*).

To test this requirement, we examined the effects of weakening the negative feedback. We created a stable H1299 cell line expressing Raf S289/296/301A, a Raf-1 mutant in which three serine residues that are phosphorylated by dpERK are mutated to alanine (*Dougherty et al., 2005*; *Hekman et al., 2005*). Assessing the dynamic range of the input as before (0–4 ng/mL EGF), we now found that dpERK responds nonlinearly to EGF dose (blue circles, *Figure 3D*), consistent with model predictions (grey line, *Figure 3D*). As a control, we found that overexpressing WT Raf-1 to a similar level does not perturb linearity (experiments, *Figure 3—figure supplement 5*; modeling,

*Figure 2—figure supplement 1*). Lastly, mutating all five direct ERK feedback sites on Raf-1 to alanine had a similar effect to Raf S289/296/301A (*Figure 3—figure supplement 6*). Our results support the model requirement that strong negative feedback is critical to linear signal transmission in the ERK pathway.

## Discussion

Our study suggests that the canonical Wnt pathway, the ERK pathway, and the Tgfβ pathway have converged upon a shared strategy of linear signal transmission. Our mathematical analysis reveals that, despite their distinct architectures, the three signaling pathways behave in some physiological contexts as linear transmitters. Not only is linearity is predicted within measured parameter regimes, the analysis shows that linearity is a property of the systems that occurs through a considerable range of parameters (*Figure 2—figure supplements 5* and *6*). We then showed direct measurements of the linear input-output relationship in the canonical Wnt and ERK pathway.

It would be interesting to further probe the generality of linear signal transmission. Linear behavior requires that single cells responds to ligand in a graded manner. Although there are reports of oscillatory or bimodality in signaling pathways, there are also multiple observations across biological contexts of single cells responding to ligand in a graded manner (*Appendix 1—table 4*). Besides the systems analyzed here, NF-κB is another signaling pathway that has been modeled rigorously (*Hoffmann et al., 2002*; *Ashall et al., 2009*; *Lee et al., 2014*). Numerical simulations of a well-established NF-κB model (*Ashall et al., 2009*) over the range of nuclear NF-κB translocation observed in human epithelial cells (*Lee et al., 2014*) reveal that the peak of the nuclear NF-κB pulse correlates linearly with ligand concentration (*Figure 2—figure supplement 7*). Finally, linearity extends beyond metazoan signaling pathways. In the yeast pheromone sensing pathway, a homolog of the ERK cascade, transcriptional output correlates linearly with receptor occupancy (*Yu et al., 2008*). The linearity is mediated by negative feedback by Fus3 acting on Sst2, a feedback that is not conserved in the mammalian ERK system. These further argue for linear signal transmission as a convergent property across independently evolving signaling pathways, as well as between conserved pathways that diverged 1.5 billion years ago.

What are potential advantages to linear signal transmission? Linearity is a feature of many engineering systems, where it serves several practical purposes. In particular, linear signal transmission enables the superposition of multiple signals, where the output of two simultaneous inputs is equal to the sum of the outputs for each input separately. Superposition enables multiple, dynamic signals to be faithfully transmitted and processed independently. Thus, for instance, linearity enables people to listen to a phone call and interpret speech amongst background noise, and allows a car radio to tune into one station out of multiple broadcasting on separate carrier frequencies. Notably, linearity is also a desired goal in synthetic biology, where it is often implemented using negative feedback (*Nevozhay et al., 2009*; *Del Vecchio et al., 2016*). Analogous to engineered circuits, linearity in biological signaling pathways may facilitate multiplexing inputs into a single pathway (*Figure 4A*).

A second benefit is that linearity might underlie two phenomena that are increasingly found across signaling pathways. First, a linear transmitter naturally gives rise to dose-response alignment (*Andrews et al., 2016*), where one or more downstream responses of a pathway closely follows the fraction of occupied receptor (*Figure 4B*). Dose response alignment appears in many biological systems and is thought to improve the fidelity of information transfer through signaling pathways (*Oyarzún et al., 2014*; *Yu et al., 2008*; *Andrews et al., 2016*; *Becker et al., 2010*). Second, linearity facilitates fold change detection, where cells sense fold changes in signal, rather than absolute level, to buffer cellular noise (*Goentoro and Kirschner, 2009*; *Cohen-Saidon et al., 2009*; *Lee et al., 2014*; *Thurley et al., 2014*; *Frick et al., 2017*). In linear input-output systems, the stimulated output correlates linearly to the basal output; thus, the fold-change in output is robust to variations in cellular parameters (*Figure 4C*). Indeed, for the signaling pathways studied here, it has been shown experimentally that the robust outcome of ligand stimulation is the fold-change in the level of transcriptional regulator (*Goentoro and Kirschner, 2009*; *Cohen-Saidon et al., 2009*; *Lee et al., 2014*; *Frick et al., 2017*). Therefore, selecting for linearity may naturally confer the benefits of superposition, dose-response alignment, and a robust fold-change in output.

Interestingly, unlike synthetic circuits whose linearity is often designed to extend across multiple orders of magnitude (*Nevozhay et al., 2009*; *Nevozhay et al., 2013*), the linearity we observed in

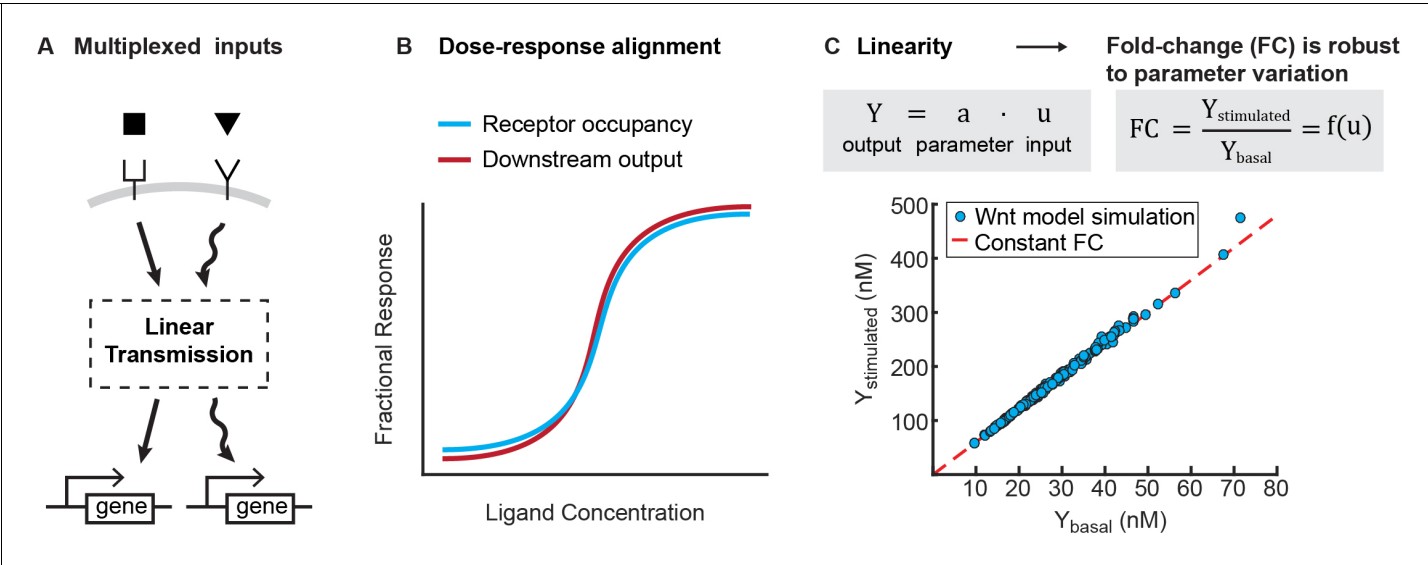

**Figure 4.** Benefits of linearity. (**A**) Linearity enables multiplexing of inputs to a signaling pathway. Multiplexed signals can be independently decoded downstream, and therefore regulate distinct transcriptional events. (**B**) Illustration for how linearity between the receptor occupancy and downstream outputs gives rise to dose-response alignment (*Andrews et al., 2016*). (**C**) Linearity can produce fold-changes in output that are robust to variation in cellular parameters. To illustrate this, we added lognormal noise (0.1 CV) to all parameters of the Wnt model, and simulated the level of β-catenin before and after Wnt stimulation (blue circles). As long as the model operates in the regime of linear signal transmission (i.e. $Y = a \cdot u$, where $Y$ is output, $u$ is input, and $a$ is a scalar that is a function of parameters), variation in parameters affects stimulated and basal level of β-catenin equally, and we get a constant fold change in β-catenin (i.e. red line, where $FC = Y_{stimulated}/Y_{basal}$ is independent of parameter variations).
DOI: https://doi.org/10.7554/eLife.33617.023

the three natural pathways extends only one order of magnitude, which is also the dynamic range of the pathways. However, we know that natural pathways can convey inputs varying across multiple orders of magnitude, for example, vision. Thus, an advantage of linearity in natural pathways may be that, in conjunction with fold-change detection at the receptor-level (*Olsman and Goentoro, 2016*), the system as a whole can continually adapt to a given input, hence maintaining sensitivity to future signals.

Why evolve complexity in signaling pathways only to produce seemingly simple behavior? We offer two thoughts. First, complexity of each pathway might afford tunability, in the sense that parameters can be tuned to produce different behaviors in different contexts. For instance, the ERK pathway produces digital, all-or-none response in some contexts (*Huang and Ferrell, 1996*), and analog response in others (*Whitehurst et al., 2004*; *Mackeigan et al., 2005*). Second - to take an example from engineering - in order to utilize physical processes that are not naturally linear, engineers must implement complex design features to approximate linearity. Similarly, many biochemical processes are inherently nonlinear, meaning that linearity does not arise from a reduction in complexity. Indeed, in each pathway we analyzed here, linearity emerges *from* complex interactions: a futile cycle in the Wnt pathway, ultrasensitivity coupled to feedback in the ERK pathway, and continual nucleocytoplasmic shuttling in the Tgfβ pathway. Therefore, analogous to engineered systems, complexity in the biochemical pathways we analyzed here might have evolved in part to produce linearity.

## Materials and methods

### Expression constructs
pBABEpuro-CRAF that contains the wt human Raf-1 clone was a gift from Matthew Meyerson (Addgene plasmid # 51124). Mutant Raf (S289/296/301A) and (S29/289/296/301/642A) were generated using the Q5 site-directed mutagenesis kit (New England Biolabs, E0554S). The mutant and wt Raf-1 were then placed downstream of a CMV promoter.

## Cell lines and cell culture

RKO cells (ATCC, CRL-2577) and H1299 cells (ATCC, CRL-5803) were authenticated by STR profiling and supplied by ATCC. RKO cells were cultured at 37°C and 5% (vol/vol) CO2 in DMEM (Thermo-Fisher Scientific; 11995) supplemented with 10% (vol/vol) FBS (Invitrogen; A13622DJ), 100 U/mL penicillin, 100 μg/mL streptomycin, 0.25 μg/mL amphotericin, and 2 mML-glutamine (Invitrogen). H1299 cells were cultured at 37C and 5% (vol/vol) CO2 in RPMI (ThermoFisher Scientific; 11875) supplemented with 10% (vol/vol) FBS (Invitrogen; A13622DJ), 100 U/mL penicillin, 100 μg/mL streptomycin, 0.25 μg/mL amphotericin, and 2 mML-glutamine (Invitrogen). Both cell lines tested negative for mycoplasma contamination.

## Transfection of Raf-1 constructs

H1299 cells were transfected with the mutant and wt Raf-1 constructs using Lipofectamine 3000 (ThermoFisher Scientific, L3000). Stable expression was selected using puromycin at a concentration of 1.5 μg/mL for 2 weeks.

## Reagents and antibodies

The following antibodies were purchased from Cell Signaling Technologies: anti-Phospho-p44/42 MAPK (Erk1/2) (Thr202/Tyr204) (E10) Mouse mAb #9106, anti-histone H3 (D1H2) XP Rabbit mAb #4499, anti-c-Raf Antibody #9422, anti-phospho-LRP6 (Ser1490) Antibody #2568, anti-GAPDH (D4C6R) Mouse mAb #97166. Anti-Beta-catenin mouse mAb was purchased from BD Transduction Laboratories (#610153) and anti-GAPDH rabbit antibody was purchased from Abcam (ab9485). The following fluorescent secondary antibodies were purchased from Fisher Scientific: IRDye 800CW Goat anti-Mouse IgG (926–32210) and IRDye 680LT Goat anti-Rabbit IgG (926-68021).

Recombinant human Wnt3A was purchased from Fisher Scientific (5036WN), and recombinant human EGF was purchased from Sigma (E9644). CHIR99021 was purchased from Sigma (SML1046). Halt Protease and Phosphatase Inhibitor Cocktail (100X) was purchased from Fisher Scientific (78440).

## CHIR99021 treatment

RKO cells were pre-treated with 1 μM CHIR99021 for 24 hr before adding replacement media containing 1 μM CHIR99021 and Wnt3A for 6 hr.

## Cell lysis

RKO cells at 70% confluency were scraped in PBS, pelleted, and snap-frozen, and then thawed in NP-40 lysis buffer containing Halt inhibitor cocktail. Samples were spun down, and the supernatants were transferred to Laemmli sample buffer and boiled. The samples were then run onto a Bolt 4–12% Bis-Tris Plus Gel (Thermofisher, NW04120BOX). H1299 cells at 70% confluence were scraped in NP-40 lysis buffer containing Halt inhibitor cocktail, and further lysed in Laemmli sample buffer. Samples were spun down, and the supernatants were boiled. The samples were then run onto a Novex 4–20% Tris-Glycine Mini Gel (ThermoFisher, XP04200BOX).

## Quantitative Western blots

Proteins were transferred onto nitrocellulose membranes, blocked for one hour at room temperature (RT) with blocking buffer (Odyssey Blocking Buffer (TBS) (927–50000) or 5% milk powder in TBS) and stained overnight at 4°C with primary antibody diluted in blocking buffer. The membranes were then stained with fluorescent IR secondary antibodies diluted in blocking buffer for one hour at RT. The fluorescent signal was then imaged using the LiCOR Odyssey Imager and quantified using Odyssey Application software version 3.0. The background-subtracted intensity of the protein bands were normalized to the loading control, GAPDH and/or Histone H3 (for RKO) or Histone H3 (for H1299). These values were then normalized to the reference lanes within each gel, to allow comparison across gels. For β-catenin and phospho-LRP5/6, variation in the fold-activation from experiment to experiment could artificially stretch the data along the x- and y-axis, and introduce artifacts into the relationship between phospho-LRP5/6 and β-catenin. Therefore, for Wnt3A dose responses, the data from each gel was scaled such that the mean of 80 ng/mL and 160 ng/mL samples was equal to the mean across all gels. Finally, for each antibody used in the study, we did

careful characterization of the linear range, and verified that our measurement conditions were within the linear range of the antibody. **Technical variability of Western blot quantitation**. To confirm the effects reported, we verified that quantitation of the same sample loaded in multiple lanes in a gel gives CV < 10%, and quantitation of the same sample across multiple independent gels gives CV < 10% (*Figure 3—figure supplement 7*). As further control, we verified that normalization with loading control did not produce artificial distortion of the input-output relationship: linearity was observed without normalization in cases where loading was already uniform (*Figure 3—figure supplement 8*).

### L-1 and L2-norm analysis

L1-norm analysis was performed as described in *Nevozhay et al. (2013)*. Briefly, the data is fitted with a cubic Hermite polynomial, and rescaled along the x and y axis to [0, 1]. The L1-norm is computed as the area between the polynomial fit and the diagonal. Linearity is defined in this context as L1-norm < 0.1. L2-norm analysis for Wnt pathway data was performed using a Pearson's coefficient, and L2-norm analysis for ERK pathway data was performed using the coefficient of correlation, $r^2$.

### Akaike information criterion

To score the validity of nonlinear model fits for *Figure 3D*, we used the bias-corrected Akaike Information Criterion as described in ref. (*Spiess and Neumeyer, 2010*), which assesses goodness-of-fit and model parsimony. The weighted Aikaike $w(AIC)$ provides a comparison of all considered models, which in our case is the nonlinear ERK pathway model fit and a linear fit, with the higher score indicating a more valid model.

## Acknowledgements

We would like to thank Rob Oania for providing advice on experiments, Michael Abrams, Christopher Frick, Kibeom Kim, and Noah Olsman for comments on the manuscript, and Michael Elowitz and Richard Murray for discussions on the study.

## Additional information

### Funding

| Funder | Grant reference number | Author |
|---|---|---|
| James S. McDonnell Foundation | 220020365 | Lea Goentoro |
| National Science Foundation | NSF.145863 | Lea Goentoro |
| National Institutes of Health | 5T32GM007616-37 | Harry Nunns |

The funders had no role in study design, data collection and interpretation, or the decision to submit the work for publication.

### Author contributions

Harry Nunns, Conceptualization, Data curation, Software, Investigation, Methodology, Writing—original draft, Writing—review and editing; Lea Goentoro, Conceptualization, Funding acquisition, Methodology, Writing—original draft, Writing—review and editing

### Author ORCIDs

Harry Nunns https://orcid.org/0000-0002-9669-0039
Lea Goentoro https://orcid.org/0000-0002-3904-0195

### Decision letter and Author response

Decision letter https://doi.org/10.7554/eLife.33617.034
Author response https://doi.org/10.7554/eLife.33617.035

## Additional files

### Supplementary files

• Transparent reporting form
DOI: https://doi.org/10.7554/eLife.33617.024

### Data availability

All data generated or analysed during this study are included in the manuscript and supporting files. Source data files have been provided for Figures 2 and 3.

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

## Appendix 1

DOI: https://doi.org/10.7554/eLife.33617.025

### 1. Variable elimination

We use a variable elimination technique from *Feliu and Wiuf (2012)* to derive analytic expressions for the steady-states of the Tgfβ and ERK pathways. This technique was developed to handle the complexity of large chemical reaction networks. By eliminating variables from the steady-state solution, we can express the steady-state of the system in terms of a smaller subset of variables. This is a useful tool for analyzing the Tgfβ and ERK models, as the steady-state solution consists of a large set of variables, each with a polynomial equation describing its steady-state.

The technique works as follows: if we can identify **a cut set** within the reaction network, we can reduce the system to a set of first-order homogeneous equations with respect to that cut. This set of equations can then be solved using linear algebra.

A cut is a set of species such that for every reaction involving those species, there is exactly one reactant and one product that falls within that cut. For example, let us consider a network of four interacting species, A, B, C, and D.

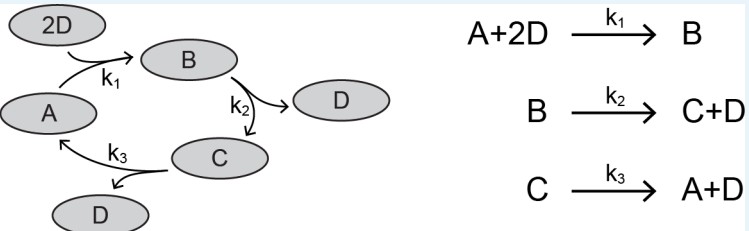

**Appendix 1—scheme 1.** Network of four proteins.

DOI: https://doi.org/10.7554/eLife.33617.026

In this network, there is a cut $\{A, B, C\}$ that contains exactly one product and one reactant for each reaction. We have highlighted this cut in the reaction set:

$$A+2D \xrightarrow{k_1} B$$

$$B \xrightarrow{k_2} C+D$$

$$C \xrightarrow{k_3} A+D$$

**Appendix 1—scheme 2.** Reaction set corresponding to protein network.

DOI: https://doi.org/10.7554/eLife.33617.027

The species D cannot belong in the cut, since it appears twice as a reactant in the first reaction. The behavior of this network is described by four differential equations,

$$[\dot{A}] = -k_1[A] \cdot [D]^2 + k_3[C] = 0 \tag{A1}$$

$$[\dot{B}] = k_1[A] \cdot [D]^2 - k_2[B] = 0 \tag{A2}$$

$$[\dot{C}] = k_2[B] - k_3[C] = 0 \tag{A3}$$

$$[\dot{D}] = -2k_1[A] \cdot D^2 + k_2[B] + k_3[C] = 0 \tag{A4}$$

which are set to zero at steady-state, and two additional conservation equations:

$$T_1 = [A] + [B] + [C] \tag{A5}$$

$$T_2 = 2[B] + [C] + [D] \tag{A6}$$

The variable elimination technique allows us to reduce the steady-state system of equations by four (three equations for the cut set, and one conservation equation). We do this by expressing each member of the cut set as a dependent variable of D, shown below. We utilize the fact that the differential equations for A, B, and C are first-order and homogenous with respect to our cut, and rewrite them in matrix form. We use the subscript 'ss' to denote steady-state:

$$\begin{pmatrix} -k_1[D]_{ss}^2 & 0 & k_3 \\ k_1[D]_{ss}^2 & -k_2 & 0 \\ 0 & k_2 & -k_3 \end{pmatrix} \begin{pmatrix} c[A]_{ss} \\ [B]_{ss} \\ [C]_{ss} \end{pmatrix} = 0 \tag{A7}$$

*Feliu and Wiuf (2012)* provides a proof of why a cut set guarantees that we can rewrite the corresponding equations in matrix form. It can be understood intuitively from the fact that a cut contains exactly one reactant of each reaction, and therefore each rate is first-order with respect to the cut. Homogeneity also follows from this, since there are no rate terms that do not include members of the cut.

For a complex model, there is no guarantee that we can derive closed-form analytical solutions for steady-state. The matrix formulation and variable elimination technique immediately provides us with a set of solvable variables. The solution to the matrix equation above is:

$$[A]_{ss} = c \cdot k_2 k_3 \tag{A8}$$

$$[B]_{ss} = c \cdot k_1 k_3 D_{ss}^2 \tag{A9}$$

$$[C]_{ss} = c \cdot k_1 k_2 D_{ss}^2 \tag{A10}$$

$c$ is a scaling factor not constrained by the matrix equation. With the use of the conservation Equation 5, we can calculate $c$ and express the steady state of all three species solely in terms of the parameters of the network, and $[D]_{ss}$. For instance, the solution for $[C]_{ss}$ is below.

$$[C]_{ss} = \frac{k_1 k_2 [D]_{ss}^2}{k_2 k_3 + k_1 [D]_{ss}^2 (k_2 + k_3)} T_1 \tag{A11}$$

The solutions for $[A]_{ss}$, $[B]_{ss}$, and $[C]_{ss}$ derived from the variable elimination technique still depend on $[D]_{ss}$. If we plug in the solutions for the cut species, we can obtain polynomial equations for the remaining species (in this case $[D]_{ss}$), but closed form expressions are not necessarily obtainable. In all the cases analyzed in this paper, variables that appear in the analytical solutions for the cut set happen to be approximately constant across a wide range of input values, as they are present in excess relative to other species.

Finally, each parameter group is physically meaningful. For instance, $k_2 k_3$, $k_1 k_3 [D]_{ss}^2$, and $k_1 k_2 [D]_{ss}^2$ represent the un-normalized fraction of $T_1$ that exists as A, B, and C, respectively. The normalization factor for these fractions is $c/T_1$, or in this case, simply the sum of all parameter groups. This provides an intuitive way of analyzing how parameter groups affect the overall distribution of $T_1$. For instance, increasing the value of $k_1$ will increase the amount of $T_1$ that exists as B and C, while necessarily decreasing the amount of A (assuming $[D]_{ss}$ does not change significantly).

## 2. Wnt model

We analyzed a mathematical model of the canonical Wnt pathway built by *Lee et al., 2003*. The model is illustrated in *Figure 2A*, and consists of 7 ODEs and 22 parameters, reproduced in *Appendix 1—table 1*.

## Solving the Wnt model at steady-state

We previously derived an expression for β-catenin in steady-state (*Goentoro and Kirschner, 2009*):

$$[\beta\text{cat}]_{ss} = K_{17} \frac{1 - \gamma + \frac{\alpha}{u(\text{Wnt})}}{2} \left( \sqrt{1 + \frac{4\gamma}{\left(1 - \gamma + \frac{\alpha}{u(\text{Wnt})}\right)^2}} - 1 \right) \tag{A12}$$

where the parameters are dimensionless groups of the binding rate constants and protein concentrations:

$$\alpha = \frac{k_4 \cdot k_6 \cdot k_9 \cdot v_{14} \cdot \text{GSK3}_{tot} \cdot \text{APC}_{tot}}{k_5 \cdot k_{-6} \cdot K_7 \cdot K_8 \cdot k_{13} \cdot k_{15}} \tag{A13}$$

$$\gamma = \frac{v_{12}}{k_{13} \cdot K_{17}} \tag{A14}$$

$$u(\text{Wnt}) = 1 + \frac{k_3 \cdot \text{Dvl}_{tot}}{k_{-6}} \cdot \frac{k_1 \cdot \text{Wnt}}{k_2 + k_1 \cdot \text{Wnt}} \tag{A15}$$

The input function $u = u(\text{Wnt})$ corresponds to the rate at which Wnt stimulation inhibits the destruction complex, normalized by $k_{-6}$. The value of Wnt ranges from 0 to 1 in the model. Please refer to *Goentoro and Kirschner (2009)* for the physical intuition of each parameter group.

## Derivation of linear behavior

We calculate the value of the parameter groups, as well as the value of the input function at saturating Wnt stimulation:

$$\alpha = 66$$

$$\gamma = 1.4$$

$$u(\text{Wnt} = 1) = 6.0$$

Within the parameter regime measured in cells, the analytical expression for β-catenin dramatically simplifies. We can perform the following first-order Taylor expansion:

$$\sqrt{1 + \epsilon} \approx 1 + \frac{1}{2}\epsilon, \quad \epsilon \ll 1 \tag{A16}$$

$$\epsilon = \frac{4\gamma}{\left(1 - \gamma + \frac{\alpha}{u}\right)^2} \tag{A17}$$

This holds true for $\frac{\alpha}{u} \gg \gamma$. Furthermore, we can make the approximation $1 - \gamma + \frac{\alpha}{u} \approx \frac{\alpha}{u}$ as long as $\frac{\alpha}{u} \gg 1$ also holds. We can encompass these two inequalities within $\alpha/u \gg 1 + \gamma$. The equation simplifies to:

$$[\beta\mathrm{cat}]_{ss} \approx \mathrm{K}_{17}\frac{\gamma}{\alpha}\mathrm{u} \tag{A18}$$

## 3. ERK model

We analyzed a mathematical model built by *Huang and Ferrell (1996)*, and revised by *Sturm et al. (2010)*. The model is illustrated in *Figure 2B*, and contains 26 ODEs and 46 parameters, reproduced in *Appendix 1—table 2*.

We changed two parameters from the original model, which are shown in *Appendix 1—table 2*. $\mathrm{k}_{25}$ characterizes the negative feedback from dpERK to unphosphorylated Raf, and $\mathrm{k}_{27}$ characterizes the negative feedback from dpERK to phosphorylated Raf. In *Sturm et al. (2010)*, the values of these parameters were estimated, rather than measured. Experimental measurements indicate that dpERK mostly interacts with Raf, and that this feedback causes strong repression of Raf (*Dougherty et al., 2005*). We therefore increased the value of $\mathrm{k}_{25}$, and set $\mathrm{k}_{27}$ to zero.

### Solving the ERK model at steady-state

In the ERK pathway, doubly phosphorylated ERK is produced by the Raf/MEK/ERK cascade of phosphorylation,

$$[\mathrm{dpERK}]_{ss} = \mathrm{g}\left([\mathrm{pRaf}]_{ss}\right) \tag{A19}$$

There is a negative feedback within the pathway, such that,

$$[\mathrm{pRaf}]_{ss} = \mathrm{f}\left(\mathrm{u}, [\mathrm{dpERK}]_{ss}\right) \tag{A20}$$

where $\mathrm{u}$ is the input function, the concentration of RasGTP (a function of ligand dose).

We first focus on deriving the negative feedback function in *Equation A20*. Using the variable elimination techniques in section 'Variable Elimination', we identify the following cut set:

$$\{\mathrm{Raf, Raf:RasGTP, pRaf, pRaf:P1, MEK:pRaf, pMEK:pRaf, Raf:ppERK, Rafi, Rafi:P4}\}$$

This allows us to express the steady-state concentration of pRaf as a function of parameters, and the remaining species in the ERK pathway. Specifically, members of this cut interact directly with, and have dependencies on, the following set:

$$\{\mathrm{P1, MEK, pMEK, dpERK, P4}\}$$

With this, we derive the expression for $[\mathrm{pRaf}]_{ss}$,

$$[\mathrm{pRaf}]_{ss} = \frac{\alpha \cdot \mathrm{u}}{\beta \cdot [\mathrm{dpERK}]_{ss} + (\alpha + \gamma) \cdot \mathrm{u} + \delta} \cdot \mathrm{Raf}_{tot} \tag{A21}$$

where the parameter groups are:

$$\alpha = \frac{\mathrm{k}_3 \cdot (\mathrm{k}_8 + \mathrm{k}_{b7})}{\mathrm{k}_7 \cdot [\mathrm{P1}]_{ss} \cdot \mathrm{k}_8} + \cdots \tag{A22}$$

$$\beta = \frac{\mathrm{k}_{25} \cdot (\mathrm{k}_{30} + \mathrm{k}_{b29} + \mathrm{k}_{29} \cdot [\mathrm{P4}]_{ss})}{\mathrm{k}_{29} \cdot [\mathrm{P4}]_{ss} \cdot \mathrm{k}_{30}} + \cdots \tag{A23}$$

$$\gamma = \frac{\mathrm{k}_3 \cdot (\mathrm{k}_8 + \mathrm{k}_{b7}) \cdot (\mathrm{k}_9 \cdot [\mathrm{MEK}]_{ss})}{\mathrm{k}_7 \cdot [\mathrm{P1}]_{ss} \cdot \mathrm{k}_8 \cdot \mathrm{k}_{10}} + \cdots \tag{A24}$$

$$\delta = \frac{k_{26} + k_{b25}}{k_{26}} + \ldots \tag{A25}$$

The ellipses indicate additional small terms (i.e. <10% of the previous terms, numerically calculated using the model parameters and $u = 4.5e4$ molecules). All the calculations for this paper use these truncated parameter groups. The complete parameter groups are written below:

$$\alpha = \ (k_3 \cdot (k_8 + k_{b7}) \cdot (k_{10} + k_{b9}) \cdot (k_{12} + k_{b11}) \cdot (k_{26} + k_{b25})) / \\ ([P1]_{ss} \cdot k_7 \cdot k_8 \cdot k_{10} \cdot k_{12} \cdot k_{26})$$

$$\beta = \ (k_{25} \cdot (k_4 + k_{b3}) \cdot (k_{10} + k_{b9}) \cdot (k_{12} + k_{b11}) \cdot (k_{26} \cdot k_{30} + k_{26} \cdot k_{b29} + [P4]_{ss} \cdot k_{26} \\ \cdot k_{29} + [P4]_{ss} \cdot k_{29} \cdot k_{30})) / ([P4]_{ss} \cdot k_4 \cdot k_{10} \cdot k_{12} \cdot k_{26} \cdot k_{29} \cdot k_{30})$$

$$\gamma = \ (k_3 \cdot (k_{26} + k_{b25}) \cdot (k_4 \cdot k_8 \cdot k_{10} \cdot k_{12} + k_4 \cdot k_8 \cdot k_{10} \cdot k_{b11} + k_4 \cdot k_8 \cdot k_{12} \cdot k_{b9} + k_4 \\ \cdot k_{10} \cdot k_{12} \cdot k_{b7} + k_4 \cdot k_8 \cdot k_{b9} \cdot k_{b11} + k_4 \cdot k_{10} \cdot k_{b7} \cdot k_{b11} + k_4 \cdot k_{12} \cdot k_{b7} \\ \cdot k_{b9} + k_4 \cdot k_{b7} \cdot k_{b9} \cdot k_{b11} + [MEK]_{ss} \cdot k_4 \cdot k_8 \cdot k_9 \cdot k_{12} + [MEK]_{ss} \cdot k_4 \cdot k_8 \\ \cdot k_9 \cdot k_{b11} + [MEK]_{ss} \cdot k_4 \cdot k_9 \cdot k_{12} \cdot k_{b7} + [MEK]_{ss} \cdot k_4 \cdot k_9 \cdot k_{b7} \cdot k_{b11} \\ + [P1]_{ss} \cdot k_4 \cdot k_7 \cdot k_{10} \cdot k_{12} + [P1]_{ss} \cdot k_7 \cdot k_8 \cdot k_{10} \cdot k_{12} + [P1]_{ss} \cdot k_4 \cdot k_7 \cdot k_{10} \\ \cdot k_{b11} + [P1]_{ss} \cdot k_4 \cdot k_7 \cdot k_{12} \cdot k_{b9} + [P1]_{ss} \cdot k_7 \cdot k_8 \cdot k_{10} \cdot k_{b11} + [P1]_{ss} \cdot k_7 \\ \cdot k_8 \cdot k_{12} \cdot k_{b9} + [P1]_{ss} \cdot k_4 * k_7 \cdot k_{b9} \cdot k_{b11} + [P1]_{ss} \cdot k_7 \cdot k_8 \cdot k_{b9} \cdot k_{b11} \\ + k_4 \cdot k_8 \cdot k_{10} \cdot k_{11} \cdot [pMEK]_{ss} + k_4 \cdot k_8 \cdot k_{11} \cdot k_{b9} \cdot [pMEK]_{ss} + k_4 \cdot k_{10} \\ \cdot k_{11} \cdot k_{b7} \cdot [pMEK]_{ss} + k_4 \cdot k_{11} \cdot k_{b7} \cdot k_{b9} \cdot [pMEK]_{ss})) / \\ ([P1]_{ss} \cdot k_4 \cdot k_7 \cdot k_8 \cdot k_{10} \cdot k_{12} \cdot k_{26})$$

$$\delta = \ ((k_4 + k_{b3}) \cdot (k_{10} + k_{b9}) \cdot (k_{12} + k_{b11}) \cdot (k_{26} + k_{b25})) / (k_4 \cdot k_{10} \cdot k_{12} \cdot k_{26})$$

## Physical significance of parameter groups

Next, we would like to develop an intuition for the physical significance of these parameter groups. As discussed in section 1, $\alpha \cdot u$ relates to the amount of free, phosphorylated Raf since $\alpha \cdot u / ((\alpha + \gamma) \cdot u + \beta[dpERK]_{ss} + \delta)$ is the fraction of Raf present as pRaf. Thus, as $\alpha \cdot u$ increases relative to $\gamma \cdot u + \beta[dpERK]_{ss} + \delta$, the amount of pRaf also increases.

We can define three subpopulations of Raf: Raf inhibited by dpERK, $[R_i]$; Raf activated by RasGTP (input), $[R_a]$; and unphosphorylated Raf $[R_n]$. Specifically:

$$[R_i] = [Raf:dpERK] + [Rafi] + [Raf:P4] \tag{A26}$$

$$[R_a] = [pRaf] + [pRaf:P1] + [MEK:pRaf] + [pMEK:pRaf] + Raf:RasGTP] \tag{A27}$$

$$[R_n] = [Raf] \tag{A28}$$

We can calculate the steady-state of each subpopulation as:

$$[R_i]_{ss} = \frac{\beta \cdot [dpERK]_{ss}}{((\alpha + \gamma) \cdot u + \beta[dpERK]_{ss} + \delta)} Raf_{tot} \tag{A29}$$

$$[R_a]_{ss} = \frac{\gamma \cdot u}{((\alpha + \gamma) \cdot u + \beta[dpERK]_{ss} + \delta)} Raf_{tot} + [pRaf]_{ss} \tag{A30}$$

$$[R_n]_{ss} = \frac{\delta}{((\alpha + \gamma) \cdot u + \beta[dpERK]_{ss} + \delta)} \cdot Raf_{tot} \tag{A31}$$

Thus, in the same sense that $\alpha \cdot u$ relates to the amount of free phosphorylated Raf, $\beta \cdot [dpERK]_{ss}$ relates to the amount of inhibited Raf, $\gamma \cdot u$ relates to the amount of

phosphorylated Raf bound to other proteins (not free), and $\delta$ relates to the amount of unphosphorylated Raf.

## Derivation of linear behavior

Now that we have derived the negative feedback function from *Equation A20*, we examine *Equation A19*. The relationship $[\text{dpERK}]_{ss} = g([\text{pRaf}]_{ss})$ is analytically intractable, because of the complexity of the phosphorylation cascade. But we know from simulations and experimental observations that it is an ultrasensitive function. From simulations, we find that a 1.3-fold change in pRaf leads to a 9-fold change in $\text{dpERK}$ (from 10% to 90% of max, *Figure 2—figure supplement 1B-C*).

We therefore approximate $[\text{pRaf}]_{ss}$ by a value $R_s$ within this range, as indicated by the dashed line in *Figure 2—figure supplement 1B*. Substituting this into the equation above and rearranging, we find that $[\text{dpERK}]_{ss}$ becomes a linear function of input:

$$[\text{dpERK}]_{ss} \approx \frac{\alpha}{\beta} \cdot \left( \frac{\text{Raf}_{tot}}{R_s} - 1 - \frac{\gamma}{\alpha} \right) \cdot u - \frac{\delta}{\beta} \tag{A32}$$

Lastly, we write the value of two terms in *Equation A32* below, numerically calculated using the parameter values of the model:

$$\frac{\alpha}{\beta} \cdot \frac{\text{Raf}_{tot}}{R_s} = 140$$

$$\frac{\alpha}{\beta} \left( 1 + \frac{\gamma}{\alpha} \right) = 13$$

We can neglect the second term, yielding:

$$[\text{dpERK}]_{ss} \approx \frac{\alpha_1}{\beta} \cdot \frac{\text{Raf}_{tot}}{R_s} \cdot u - \frac{\delta}{\beta} \tag{A33}$$

## Derivation for treating pRaf as a constant

Next, we analyze exactly how the level of pRaf changes with the input u. From earlier, we have that

$$[\text{dpERK}]_{ss} = g([\text{pRaf}]_{ss}) \tag{A34}$$

$$[\text{pRaf}_{tot}]_{ss} = f(u, [\text{dpERK}]_{ss}) \tag{A35}$$

We can now derive a general expression for the relative change of $[\text{pRaf}]_{ss}$ with respect to a relative change in u. We use the notation $dx = d\ln x = dx/x$.

$$\frac{d\hat{f}}{d\hat{u}} = \frac{\partial f}{\partial u} \cdot \frac{u}{f} \cdot \left( 1 - \frac{\partial f}{\partial [\text{dpERK}]_{ss}} \cdot \frac{dg}{d[\text{pRaf}]_{ss}} \right)^{-1} \tag{A36}$$

Next, we define the response coefficient K between $[\text{dpERK}]_{ss}$ and $[\text{pRaf}]_{ss}$:

$$K \triangleq \frac{dg}{d[\text{pRaf}]_{ss}} \cdot \frac{[\text{pRaf}]_{ss}}{[\text{dpRaf}]_{ss}} \tag{A37}$$

From *Equation A21*, we get the partial derivatives:

$$\frac{\partial f}{\partial u} = \frac{f}{u} \cdot \frac{\beta[\text{dpERK}]_{ss} + \delta}{\beta[\text{dpERK}]_{ss} + (\alpha + \gamma)u + \delta} \tag{A38}$$

$$\frac{\partial f}{\partial [\text{dpERK}]_{ss}} = -f \cdot \frac{\beta}{\beta [\text{dpERK}]_{ss} + (\alpha + \gamma) u + \delta} \tag{A39}$$

Using these two equations, we find that:

$$\frac{d\hat{f}}{d\hat{u}} = \frac{\beta [\text{dpERK}]_{ss} + \delta}{(1+k) \cdot \beta [\text{dpERK}]_{ss} + \alpha u + \delta} \tag{A40}$$

When $K \gg 1$ and $\beta [\text{dpERK}]_{ss} \sim (\alpha + \gamma) \cdot u + \delta$, we see that

$$\frac{d\hat{f}}{d\hat{u}} \approx K^{-1} \tag{A41}$$

Therefore, $[\text{pRaf}]_{ss}$ is held constant in the region where the kinase cascade is ultrasensitive and feedback is strong. In this region, it is easy to show that $[\text{dpERK}]_{ss}$ becomes a linear function of input.

$$\frac{d\hat{g} - g_0}{d\hat{u}} \approx 1; \qquad g_0 = -\frac{\delta}{\beta} \tag{A42}$$

It is not guaranteed that the system is stable as $K$ increases, but we see from simulations that our parameter regime provides a stable output.

## Toy model of the ERK pathway

Here we utilize a toy model to illustrate how ultrasensitivity and strong negative feedback combine to generate input-output linearity. In this model, induction of the output species $E$ is a two-step process:

1. An input $u$ increases the amount of species $R$, which in turn influences $E$ as $E = g(R)$. There is negative feedback from $E$ to $R$, which in the limit of strong negative feedback is inversely proportional to $E$.
2. Next, we specify the function $g(R)$ such that $K = K_0$, where $K$ is the relative change of $E$ with respect to $R$. As $K_0$ increases, therefore, the function $g(R)$ becomes more ultrasensitive.

Solving for $E$, we see that in the limit of $K = K_0 \gg 1$, $E$ becomes a linear function of $u$, and $R$ is held constant at $R_s$.

While we do not have an explicit function for $g(R)$ for the full ERK model, we include derivations in section "**Derivation for treating pRaf as a constant**" that show that these results hold for any function $g(R)$ in the region where $K \gg 1$. We also show that these results hold outside the limit of strong negative feedback, as long as the feedback-inhibited pool of $R$ is comparable to the remaining pool.

**A**

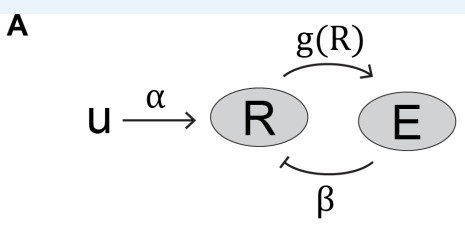

**strong feedback**

$$R = f(u,E) = \frac{\alpha \cdot u}{\beta \cdot E}$$

**g(R) such that K = $K_0$**

$$E = g(R) = \begin{cases} E_s \cdot (R/R_s)^{K_0} & \text{for } R \leq R_s \\ E_s & \text{for } R > R_s \end{cases}$$

**input-output response**

$$E = E_s^{\frac{1}{1+K_0}} \left( \frac{\alpha \cdot u}{\beta \cdot R_s} \right)^{\frac{K_0}{1+K_0}}$$

$$\lim_{K \to \infty} E = \frac{\alpha \cdot u}{\beta \cdot R_s}$$

**B**

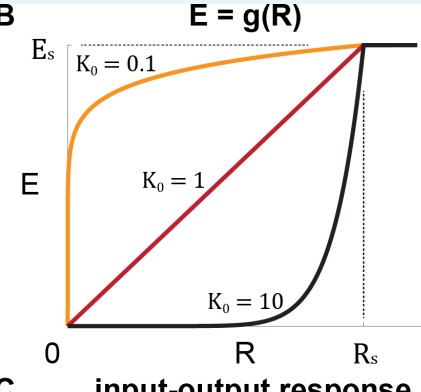

**C**

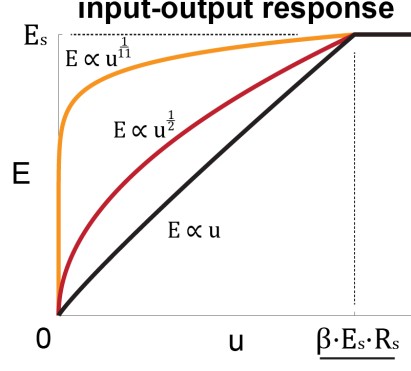

**Appendix 1—scheme 3.** Toy model of the ERK pathway.
DOI: https://doi.org/10.7554/eLife.33617.028

## 4. Tgfβ model

We analyzed a mathematical model built by *Schmierer et al. (2008)*. The model is illustrated in *Figure 2C*, and consists of 10 ODEs and 14 parameters, reproduced in *Appendix 1—tables 3*.

### Solving the Tgfβ model at steady-state

We use the variable elimination technique described in section 'Variable Elimination' to derive an analytical expression for the steady-state concentration of nuclear Smad complex. First, based on the measured parameter values, and as confirmed by simulations, the extent of Smad2-Smad2 binding is limited. We therefore neglect this reaction in subsequent analysis. We identify the following cut of the Tgfβ model:

$$\{S2c, pS2c, S24c, S2n, pS2n, S24n\}$$

which is subject to the conservation equation:

$$([S2c] + [pS2c] + [S24c]) + \frac{1}{a}([S2n] + [pS2n] + [S24n]) = S2_{tot} \tag{A43}$$

Thus, we can eliminate these variables from the steady-state polynomial solution, with dependence only on variables outside this cut:

$$\{S4c, S4n\}$$

Using this relationship, we derive an expression for the nuclear Smad complex(S24n) at steady-state,

$$[\text{S24n}]_{ss} = \frac{a \cdot \alpha \cdot u}{(\alpha + \gamma) \cdot u + \beta} \text{S2}_{tot} \tag{A44}$$

where the parameter groups are:

$$\alpha = \frac{a \cdot (k_{on}[\text{S4n}]_{ss} + a \cdot k_{ex2})}{k_{off}} + \cdots \tag{A45}$$

$$\beta = \text{PPase} \cdot \frac{k_{dephos}}{k_{phos} \cdot R_{tot} \cdot \frac{k_{ex2}}{a \cdot k_e x2} + k_{in2}} + \cdots \tag{A46}$$

$$\gamma = a \cdot (a \cdot k_{ex2} + \text{PPase} \cdot k_{dephos}) \left( \frac{1}{a \cdot k_{ex2}} + \frac{1}{\text{CIF} \cdot k_{in2}} \right) + \cdots \tag{A47}$$

Here the input function $u = u(\text{Tgf}\beta)$ is the fraction of receptors activated by Tgf$\beta$ ligands. The ellipses indicate additional small terms (i.e. <10% of the previous terms, as calculated using the model parameters, with the variables $[\text{S4c}]_{ss}$ and $[\text{S4n}]_{ss}$ calculated for $u = 0$). All calculations for the paper use these truncated parameter groups. The complete parameter groups are written below:

$$\alpha = \big( a \cdot ([\text{S4n}]_{ss} \cdot k_{off} + \text{CIF} \cdot [\text{S4n}]_{ss} \cdot k_{in2} + \text{CIF} \cdot \text{PPase} \cdot [\text{S4c}]_{ss} \cdot k_{dephos} + \text{CIF} \\ \cdot [\text{S4c}]_{ss} \cdot [\text{S4n}]_{ss} \cdot k_{on} + \text{CIF} \cdot [\text{S4c}]_{ss} \cdot a \cdot k_{ex2}) \big) / \big( \text{CIF} \cdot [\text{S4c}]_{ss} \cdot k_{off} \big)$$

$$\beta = \big( \text{PPase} \cdot k_{dephos} \cdot (k_{in2} + a \cdot k_{ex2}) \cdot (k_{off} + \text{CIF} \cdot k_{in2} + \text{CIF} \cdot [\text{S4c}]_{ss} \cdot k_{on}) \big) / \\ \big( \text{CIF} \cdot R_{tot} \cdot [\text{S4c}]_{ss} \cdot k_{ex2} \cdot k_{on} \cdot k_{phos} \big)$$

$$\gamma = \big( (\text{PPase} \cdot k_{dephos} + a \cdot k_{ex2}) \cdot (k_{in2} \cdot k_{off} + \text{CIF} \cdot k_{in2}^2 + a \cdot k_{ex2} \cdot k_{off} + \text{CIF} \cdot [\text{S4c}]_{ss} \\ \cdot k_{in2} \cdot k_{on} + \text{CIF} \cdot a \cdot k_{ex2} \cdot k_{in2} + [\text{S4c}]_{ss} \cdot a \cdot k_{ex2} \cdot k_{on}) \big) / \\ \big( \text{CIF} \cdot [\text{S4c}]_{ss} \cdot k_{ex2} \cdot k_{in2} \cdot k_{on} \big)$$

## Physical significance of parameter groups

Next, we would like to develop an intuition for the physical significance of these parameter groups. As discussed in section 1, $\alpha \cdot u$ relates to the amount of nuclear Smad complex since $\alpha \cdot u / ((\alpha + \gamma) \cdot u + \beta)$ is the fraction of Smad2 present as S24n. Thus, as $\alpha \cdot u$ increases relative to $\gamma \cdot u + \beta$, the amount of S24n also increases.

By definition, the parameter groups $\beta$ and $\gamma \cdot u$ capture the remaining input-independent and input-dependent polynomials, respectively. Nevertheless, we would like to understand the physical significance of the parameter groups. We can calculate the amount of unphosphorylated Smad2 as:

$$[\text{S2c}]_{ss} + \frac{1}{a}[\text{S2n}]_{ss} = \frac{\beta + \delta \cdot u}{\beta + (\alpha + \gamma) \cdot u} \text{S2}_{tot} \tag{A48}$$

$$\delta = \text{PPase} \cdot k_{dephos} \cdot \frac{k_{off} + \text{CIF} \cdot k_{in2} + \text{CIF} \cdot [\text{S4c}]_{ss} \cdot k_{on}}{\text{CIF} \cdot [\text{S4c}]_{ss} \cdot k_{ex2} \cdot k_{on}} \tag{A49}$$

$\delta$ captures the dependence of nuclear, unphosphorylated Smad on the input. With the measured parameters, $\beta \gg \delta \cdot u$, so we have

$$[\text{S2c}]_{ss} + \frac{1}{a}[\text{S2n}]_{ss} \approx \frac{\beta}{\beta + (\alpha + \gamma) \cdot u} \text{S2}_{tot} \tag{A50}$$

This means that $\beta$ relates to the amount of unphosphorylated Smad2 in the same sense that $\alpha \cdot u$ relates to nuclear Smad complex. We can also express the remaining Smad2 species as:

$$[pS2c]_{ss} + [S24c]_{ss} + \frac{1}{a}[pS2n]_{ss} = \frac{(\gamma - \delta) \cdot u}{((\alpha + \gamma) \cdot u + \beta)} \cdot S2_{tot} \qquad (A51)$$

However, as $\delta$ is of the same order of magnitude as $\gamma$, the parameter group $\gamma$ only loosely relates to these remaining species of Smad2.

## Derivation of linear behavior

Within the parameter values measured in cells, the behavior of Smad complex dramatically simplifies. Using the measured values (*Appendix 1—table 3*), the parameter groups are

$$\alpha \cdot u = 3.1$$

$$\gamma \cdot u = 1.3$$

$$\beta = 46$$

where we have used a non-saturating input ($u = 0.2$). Therefore, with measured parameters, $\beta \gg (\alpha + \gamma) \cdot u$. With this, the denominator in the $[S4n]_{ss}$ equation simplifies, and the concentration of Smad complex becomes a linear function of the input:

$$[S24n]_{ss} \approx \frac{\alpha \cdot S2_{tot}}{\beta} \cdot u \qquad (A52)$$

**Appendix 1—table 1.** Parameters, variables, and equations of the Wnt model.

| Parameter | Label | Value | |
|---|---|---|---|
| Activation rate of Disheveled/Dvl by Wnt | $k_1$ | 0.182 | $min^{-1}$ |
| Inactivation rate of Dvl | $k_2$ | $1.82 \cdot 10^{-2}$ | $min^{-1}$ |
| Dissociation of destruction complex (DC) by active Dvl | $k_3$ | $5.00 \cdot 10^{-2}$ | $nM^{-1}\ min^{-1}$ |
| Phosphorylation of DC | $k_4$ | 0.267 | $min^{-1}$ |
| Dephosphorylation of DC | $k_5$ | 0.133 | $min^{-1}$ |
| Forward rate for DC binding | $k_6$ | $9.09 \cdot 10^{-2}$ | $nM^{-1}\ min^{-1}$ |
| Reverse rate for DC binding | $k_{-6}$ | 0.909 | $min^{-1}$ |
| Dissociation constant for APC:axin binding | $K_7$ | 50 | nM |
| Dissociation constant for β-catenin:DC binding | $K_8$ | 120 | nM |
| Phosphorylation rate of β-catenin | $k_9$ | 206 | $min^{-1}$ |
| Rate of phosphorylated β-catenin release from DC | $k_{10}$ | 206 | $min^{-1}$ |
| Degradation rate of phosphorylated β-catenin | $k_{11}$ | 0.417 | $min^{-1}$ |
| Synthesis rate of β-catenin | $v_{12}$ | 0.423 | $nM\ min^{-1}$ |
| Degradation rate of β-catenin | $k_{13}$ | $2.57 \cdot 10^{-4}$ | $min^{-1}$ |
| Synthesis rate of axin | $v_{14}$ | $8.22 \cdot 10^{-5}$ | $nM\ min^{-1}$ |
| Degradation rate of axin | $k_{15}$ | 0.167 | $min^{-1}$ |
| Dissociation constant for β-catenin:TCF binding | $K_{16}$ | 30 | nM |
| Dissociation constant for β-catenin:APC binding | $K_{17}$ | 1200 | nM |
| Total concentration of Disheveled | $Dvl_{tot}$ | 100 | nM |
| Total concentration of adenomatous polyposis coli | $APC_{tot}$ | 100 | nM |
| Total concentration of T-cell factor | $TCF_{tot}$ | 15 | nM |
| Total concentration of glycogen synthase kinase 3β | $GSK3_{tot}$ | 50 | nM |

*Appendix 1—table 1 continued on next page*

*Appendix 1—table 1 continued*

| Parameter | Label | Value |
|---|---|---|
| Independent Variable | | Label |
| Active Disheveled | | $X_2$ |
| APC*/axin*/GSK3 (* denotes phosphorylated) | | $X_3$ |
| APC/axin/GSK3 | | $X_4$ |
| β-catenin*/APC*/axin*/GSK3 | | $X_9$ |
| β-catenin* | | $X_{10}$ |
| β-catenin | | $X_{11}$ ($\beta$cat) |
| axin | | $X_{12}$ |
| Dependent Variable | | Label |
| Inactive Disheveled | | $X_1$ |
| GSK3 | | $X_5$ |
| APC/axin | | $X_6$ |
| APC | | $X_7$ |
| β-catenin/APC*/axin*/GSK3 | | $X_8$ |
| TCF | | $X_{13}$ |
| β-catenin/TCF | | $X_{14}$ |
| β-catenin/APC | | $X_{15}$ |
| Differential Equations | | |

$$[\dot{X_2}] = k_1 \cdot \text{Wnt} \cdot (\text{Dvl}_{\text{tot}} - [X_2]) - k_2 \cdot [X_2]$$

$$\left(1 + \frac{[X_{11}]}{K_8}\right) \cdot [\dot{X_3}] + \frac{[X_3]}{K_8} \cdot [\dot{X_{11}}] = k_4 \cdot [X_4] - k_5 \cdot [X_3] - \frac{k_9 \cdot [X_3] \cdot [X_{11}]}{K_8} + k_{10} \cdot [X_9]$$

$$[\dot{X_4}] = -(k_3 \cdot [X_2] + k_4 + k_{-6}) \cdot [X_4] + k_5 \cdot [X_3] + k_6 \cdot \text{GSK3}_{\text{tot}} \cdot \frac{K_{17} \cdot [X_{12}] \cdot \text{APC}_{\text{tot}}}{K_7 \cdot (K_{17} + [X_{11}])}$$

$$[\dot{X_9}] = \frac{k_9 \cdot [X_3] \cdot [X_{11}]}{K_8} - k_{10} \cdot [X_9]$$

$$[\dot{X_{10}}] = k_{10} \cdot [X_9] - k_{11} \cdot [X_{10}]$$

$$\left(1 + \frac{[X_3]}{K_8} + \frac{K_{16} \cdot \text{TCF}_{\text{tot}}}{(K_{16} + [X_{11}])^2} + \frac{K_{17} \cdot \text{APC}_{\text{tot}}}{(K_{17} + [X_{11}])^2}\right) \cdot [\dot{X_{11}}] + \frac{[X_{11}]}{K_8} \cdot [\dot{X_3}] = v_{12} - \left(\frac{k_9 \cdot [X_3]}{K_8} + k_{13}\right) \cdot [X_{11}]$$

$$\left(1 + \frac{K_{17} \cdot \text{APC}_{\text{tot}}}{K_7 \cdot (K_{17} + [X_{11}])}\right) \cdot [\dot{X_{12}}] - \frac{K_{17} \cdot [X_{12}] \cdot \text{APC}_{\text{tot}}}{K_7 \cdot (K_{17} + [X_{11}])^2} \cdot [\dot{X_{11}}]$$
$$= k_3 \cdot [X_2] \cdot [X_4] - k_6 \cdot \text{GSK}_{\text{tot}} \cdot \frac{K_{17} \cdot [X_{12}] \cdot \text{APC}_{\text{tot}}}{K_7 \cdot (K_{17} + [X_{11}])} + k_{-6} \cdot [X_4] + v_{14}$$
$$- k_{15} \cdot [X_{12}]$$

Equations for fast equilibrium reactions

$$[X_1] = \text{Dvl}_{\text{tot}} - [X_2]$$

$$[X_5] = \text{GSK3}_{\text{tot}}$$

$$[X_6] = \frac{K_{17} \cdot [X_{12}] \cdot \text{APC}_{\text{tot}}}{K_7 \cdot (K_{17} + [X_{11}])}$$

*Appendix 1—table 1 continued on next page*

*Appendix 1—table 1 continued*

| Parameter | Label | Value |
|---|---|---|
| $[X_7] = \frac{K_{17} \cdot APC_{tot}}{K_{17} \cdot (K_{17} + [X_{11}])}$ | | |
| $[X_8] = \frac{[X_3] \cdot [X_{11}]}{K_8}$ | | |
| $[X_{13}] = \frac{K_{16} \cdot TCF_{tot}}{K_{16} + [X_{11}]}$ | | |
| $[X_{14}] = \frac{[X_{11}] \cdot TCF_{tot}}{K_{16} + [X_{11}]}$ | | |
| $[X_{15}] = \frac{[X_{11}] \cdot APC_{tot}}{K_{17} + [X_{11}]}$ | | |

DOI: https://doi.org/10.7554/eLife.33617.029

**Appendix 1—table 2.** Parameters, variables, and equations of the ERK model. Values highlighted in yellow have been changed from the original model (explained in section 'ERK Model').

| Parameter | Label | Value | |
|---|---|---|---|
| Forward rate for Raf:RasGTP binding | $k_3$ | $1.67 \cdot 10^{-6}$ | molecule$^{-1}$s$^{-1}$ |
| Reverse rate for Raf:RasGTP binding | $k_{b3}$ | $5.3 \cdot 10^{-3}$ | s$^{-1}$ |
| Phosphorylation rate for Raf by RasGTP | $k_4$ | 1 | s$^{-1}$ |
| Forward rate of pRaf:P1 binding | $k_7$ | $1.18 \cdot 10^{-4}$ | molecule$^{-1}$s$^{-1}$ |
| Reverse rate of pRaf:P1 binding | $k_{b7}$ | 0.2 | s$^{-1}$ |
| Dephosphorylation rate of pRaf by P1 | $k_8$ | 1 | s$^{-1}$ |
| Forward rate of MEK:pRaf binding | $k_9$ | $1.95 \cdot 10^{-5}$ | molecule$^{-1}$s$^{-1}$ |
| Reverse rate of MEK:pRaf binding | $k_{b9}$ | $3.3 \cdot 10^{-2}$ | s$^{-1}$ |
| Phosphorylation rate of MEK by pRaf | $k_{10}$ | 3.5 | s$^{-1}$ |
| Forward rate of pMEK:pRaf binding | $k_{11}$ | $1.95 \cdot 10^{-5}$ | molecule$^{-1}$s$^{-1}$ |
| Reverse rate of pMEK:pRaf binding | $k_{b11}$ | $3.3 \cdot 10^{-2}$ | s$^{-1}$ |
| Phosphorylation rate of pMEK by pRaf | $k_{12}$ | 2.9 | s$^{-1}$ |
| Forward rate of dpMEK:P2 binding | $k_{13}$ | $2.38 \cdot 10^{-5}$ | molecule$^{-1}$s$^{-1}$ |
| Reverse rate of dpMEK:P2 binding | $k_{b13}$ | 0.8 | s$^{-1}$ |
| Dephosphorylation rate of dpMEK by P2 | $k_{14}$ | $5.8 \cdot 10^{-2}$ | s$^{-1}$ |
| Forward rate of pMEK:P2 binding | $k_{15}$ | $4.5 \cdot 10^{-7}$ | molecule$^{-1}$s$^{-1}$ |
| Reverse rate of pMEK:P2 binding | $k_{b15}$ | 0.5 | s$^{-1}$ |
| Dephosphorylation rate of pMEK by P2 | $k_{16}$ | $5.8 \cdot 10^{-2}$ | s$^{-1}$ |
| Forward rate of ERK:dpMEK binding | $k_{17}$ | $8.9 \cdot 10^{-5}$ | molecule$^{-1}$s$^{-1}$ |
| Reverse rate of ERK:dpMEK binding | $k_{b17}$ | $1.83 \cdot 10^{-2}$ | s$^{-1}$ |
| Phosphorylation rate of ERK by dpMEK | $k_{18}$ | 16 | s$^{-1}$ |
| Forward rate of pERK:dpMEK binding | $k_{19}$ | $8.9 \cdot 10^{-5}$ | molecule$^{-1}$s$^{-1}$ |
| Reverse rate of pERK:dpMEK binding | $k_{b19}$ | $1.83 \cdot 10^{-2}$ | s$^{-1}$ |
| Phosphorylation rate of pERK by dpMEK | $k_{20}$ | 5.7 | s$^{-1}$ |
| Forward rate of pERK:P3 binding | $k_{21}$ | $8.33 \cdot 10^{-6}$ | molecule$^{-1}$s$^{-1}$ |
| Reverse rate of pERK:P3 binding | $k_{b21}$ | 0.5 | s$^{-1}$ |

*Appendix 1—table 2 continued on next page*

Appendix 1—table 2 continued

| Parameter | Label | Value | |
|---|---|---|---|
| Dephosphorylation rate of pERK by P3 | $k_{22}$ | 0.246 | $s^{-1}$ |
| Forward rate of dpERK:P3 binding | $k_{23}$ | $2.35 \cdot 10^{-5}$ | $molecule^{-1}s^{-1}$ |
| Reverse rate of dpERK:P3 binding | $k_{b23}$ | 0.6 | $s^{-1}$ |
| Dephosphorylation rate of dpERK by P3 | $k_{24}$ | 0.246 | $s^{-1}$ |
| Forward rate of Raf:dpERK binding | $k_{25}$ | $1 \cdot 10^{-6}$ | $molecule^{-1}s^{-1}$ |
| Reverse rate of Raf:dpERK binding | $k_{b25}$ | 1 | $s^{-1}$ |
| Hyper-phosphorylation rate of Raf by ppERK | $k_{26}$ | 10 | $s^{-1}$ |
| Forward rate of pRaf:dpERK binding | $k_{27}$ | 0 | $molecule^{-1}s^{-1}$ |
| Reverse rate of pRaf:dpERK binding | $k_{b27}$ | 1 | $s^{-1}$ |
| Hyper-phosphorylation rate of phosphorylated Raf by dpERK | $k_{28}$ | 10 | $s^{-1}$ |
| Forward rate of Rafi:P4 binding | $k_{29}$ | $5 \cdot 10^{-5}$ | $molecule^{-1}s^{-1}$ |
| Reverse rate of Rafi:P4 binding | $k_{b29}$ | 0.2 | $s^{-1}$ |
| Dephosphorylation rate of Rafi by P4 | $k_{30}$ | 0.5 | $s^{-1}$ |
| Total Raf | $Raf_{tot}$ | $4 \cdot 10^4$ | molecules |
| Total MEK | $MEK_{tot}$ | $2.1 \cdot 10^7$ | molecules |
| Total ERK | $ERK_{tot}$ | $2.21 \cdot 10^7$ | molecules |
| Total phosphatase P1 | $P1_{tot}$ | $4 \cdot 10^4$ | molecules |
| Total phosphatase P2 | $P2_{tot}$ | $4 \cdot 10^5$ | molecules |
| Total phosphatase P3 | $P3_{tot}$ | $1 \cdot 10^7$ | molecules |
| Total phosphatase P4 | $P4_{tot}$ | $4 \cdot 10^4$ | molecules |
| Variable | Label | | |
| Unphosphorylated Raf | Raf | | |
| Raf bound to RasGTP | Raf:RasGTP | | |
| Phosphorylated Raf | pRaf | | |
| Phosphatase for phosphorylated Raf | P1 | | |
| Phosphorylated Raf bound to its phosphatase | pRaf:P1 | | |
| Unphosphorylated MEK | MEK | | |
| MEK bound to its kinase | MEK:pRaf | | |
| Phosphorylated MEK | pMEK | | |
| Phosphorylated MEK bound to its kinase | pMEK:pRaf | | |
| Doubly-phosphorylated MEK | dpMEK | | |
| MEK phosphatase | P2 | | |
| Doubly-phosphorylated MEK bound to its phosphatase | dpMEK:P2 | | |
| Phosphorylated MEK bound to its phosphatase | pMEK:P2 | | |
| Unphosphorylated ERK | ERK | | |
| ERK bound to its kinase | ERK:dpMEK | | |
| Phosphorylated ERK | pERK | | |
| Phosphorylated ERK bound to its kinase | pERK:dpMEK | | |
| Doubly-phosphorylated ERK | dpERK | | |
| ERK phosphatase | P3 | | |
| Phosphorylated ERK bound to its phosphatase | pERK:P3 | | |
| Doubly-phosphorylated ERK bound to its phosphatase | dpERK:P3 | | |
| Raf bound to doubly-phosphorylated ERK | Raf:dpERK | | |

Appendix 1—table 2 continued on next page

*Appendix 1—table 2 continued*

| Parameter | Label | Value |
|---|---|---|
| Hyper-phosphorylated, 'inactive' Raf | | Rafi |
| Phosphorylated Raf bound to doubly-phosphorylated ERK | | pRaf:dpERK |
| Phosphatase for hyper-phosphorylated Raf | | P4 |
| Hyper-phosphorylated Raf bound to its phosphatase | | Rafi:P4 |

Differential Equations

$$[\dot{Raf}] = -k_3 \cdot [Raf] \cdot u(EGF) + k_{b3} \cdot [Raf:RasGTP] + k_8 \cdot [pRaf:P1] - k_25 \cdot [Raf]$$
$$\cdot [dpERK] + k_{b25} \cdot [Raf:dpERK] + k_{30} \cdot [Rafi:P4]$$

$$[\dot{Raf:RasGPT}] = k_3 \cdot [Raf] \cdot u(EGF) - (k_{b3} + k_4) \cdot [Raf:RasGTP]$$

$$[\dot{pRaf}] = k_4 \cdot [Raf:RasGTP] - k_7 \cdot [pRaf] \cdot [P1] + k_{b7} \cdot [pRaf:P1] - k_9 \cdot [MEK] \cdot [pRaf]$$
$$+ (k_{b9} + k_{10}) \cdot [MEK:pRaf] - k_{11} \cdot [pMEK] \cdot [pRaf] + (k_{b11} + k_{12})$$
$$\cdot [pMEK:pRaf] - k_{27} \cdot [pRaf] \cdot [dpERK] + k_{b27} \cdot [pRaf:dpERK]$$

$$[\dot{P1}] = -k_7 \cdot [pRaf] \cdot [P1] + (k_{b7} + k_8) \cdot [pRaf:P1]$$

$$[\dot{pRaf:P1}] = k_7 \cdot [pRaf] \cdot [P1] - (k_{b7} + k_8) \cdot [pRaf:P1]$$

$$[\dot{MEK}] = -k_9 \cdot [MEK] \cdot [pRaf] + k_{b9} \cdot [MEK:pRaf] + k_{16} \cdot [pMEK:P2]$$

$$[\dot{MEK:pRaf}] = k_9 \cdot [MEK] \cdot [pRaf] - (k_{b9} + k_{10}) \cdot [MEK:pRaf]$$

$$[\dot{pMEK}] = k_{10} \cdot [MEK:pRaf] - k_{11} \cdot [pMEK] \cdot [pRaf] + k_{b11} \cdot [pMEK:pRaf] + k_{14}$$
$$\cdot [dpMEK:P2] - k_{15} \cdot [pMEK] \cdot [P2] + k_{b15} \cdot [pMEK:P2]$$

$$[\dot{pMEK:pRaf}] = k_{11} \cdot [pMEK] \cdot [pRaf] - (k_{b11} + k_{12}) \cdot [pMEK:pRaf]$$

$$[\dot{dpMEK}] = k_{12} \cdot [pMEK:pRaf] - k_{13} \cdot [dpMEK] \cdot [P2] + k_{b13} \cdot [dpMEK:P2] - k_{17} \cdot [ERK]$$
$$\cdot [dpMEK] + (k_{b17} + k_{18}) \cdot [ERK:dpMEK] - k_{19} \cdot [pERK] \cdot [dpMEK]$$
$$+ (k_{b19} + k_{20}) \cdot [pERK:dpMEK]$$

$$[\dot{P2}] = -k_{13} \cdot [dpMEK] \cdot [P2] + (k_{b13} + k_{14}) \cdot [dpMEK:P2] - k_{15} \cdot [pMEK] \cdot [P2]$$
$$+ (k_{b15} + k_{16}) \cdot [pMEK:P2]$$

$$[\dot{dpMEK:P2}] = k_{13} \cdot [dpMEK] \cdot [P2] - (k_{b13} + k_{14}) \cdot [dpMEK:P2]$$

$$[\dot{pMEK:P2}] = k_{15} \cdot [pMEK] \cdot [P2] - (k_{b15} + k_{16}) \cdot [pMEK:P2]$$

$$[\dot{ERK}] = -k_{17} \cdot [ERK] \cdot [dpMEK] + k_{b17} \cdot [ERK:dpMEK] + k_{22} \cdot [pERK:P3]$$

$$[\dot{ERK:dpMEK}] = k_{17} \cdot [ERK] \cdot [dpMEK] - (k_{b17} + k_{18}) \cdot [ERK:dpMEK]$$

$$[\dot{pERK}] = k_{18} \cdot [ERK:dpMEK] - k_{19} \cdot [pERK] \cdot [dpMEK] + k_{b19} \cdot [pERK:dpMEK] - k_{21}$$
$$\cdot [pERK] \cdot [P3] + k_{b21} \cdot [pERK:P3] + k_{24} \cdot [dpERK:P3]$$

$$[\dot{pERK:dpMEK}] = k_{19} \cdot [pERK] \cdot [dpMEK] - (k_{b19} + k_{20}) \cdot [pERK:dpMEK]$$

*Appendix 1—table 2 continued on next page*

*Appendix 1—table 2 continued*

| Parameter | Label | Value |
|---|---|---|

$$[\dot{\text{dpERK}}] = \quad \text{k}_{20} \cdot [\text{pERK} : \text{dpMEK}] - \text{k}_{23} \cdot [\text{dpERK}] \cdot [\text{P3}] + \text{k}_{\text{b23}} \cdot [\text{dpERK} : \text{P3}] - \text{k}_{25} \cdot [\text{Raf}]$$
$$\cdot [\text{dpERK}] + (\text{k}_{\text{b25}} + \text{k}_{26}) \cdot [\text{Raf} : \text{dpERK}] - \text{k}_{27} \cdot [\text{pRaf}] \cdot [\text{dpERK}]$$
$$+ (\text{k}_{\text{b27}} + \text{k}_{28}) \cdot [\text{pRaf} : \text{dpERK}]$$

$$[\dot{\text{P3}}] = \quad -\text{k}_{21} \cdot [\text{pERK}] \cdot [\text{P3}] + (\text{k}_{\text{b21}} + \text{k}_{22}) \cdot [\text{pERK} : \text{P3}] - \text{k}_{23} \cdot [\text{dpERK}] \cdot [\text{P3}]$$
$$+ (\text{k}_{\text{b23}} + \text{k}_{24}) \cdot [\text{dpERK} : \text{P3}]$$

$$[\dot{\text{pERK:P3}}] = \text{k}_{21}[\text{pERK}] \cdot [\text{P3}] - (\text{k}_{\text{b21}} + \text{k}_{22}) \cdot [\text{pERK} : \text{P3}]$$

$$[\dot{\text{dpERK:P3}}] = \text{k}_{23} \cdot [\text{dpERK}] \cdot [\text{P3}] - (\text{k}_{\text{b23}} + \text{k}_{24}) \cdot [\text{dpERK} : \text{P3}]$$

$$[\dot{\text{Raf:dpERK}}] = \text{k}_{25} \cdot [\text{Raf}] \cdot [\text{dpERK}] - (\text{k}_{\text{b25}} + \text{k}_{26}) \cdot [\text{Raf} : \text{dpERK}]$$

$$[\dot{\text{Rafi}}] = \text{k}_{26} \cdot [\text{Raf} : \text{dpERK}] + \text{k}_{28} \cdot [\text{pRaf} : \text{dpERK}] - \text{k}_{29} \cdot [\text{Rafi}] \cdot [\text{P4}] + \text{k}_{\text{b29}} \cdot [\text{Rafi} : \text{P4}]$$

$$[\dot{\text{pRaf:dpERK}}] = \text{k}_{27} \cdot [\text{pRaf}] \cdot [\text{dpERK}] - (\text{k}_{\text{b27}} + \text{k}_{28}) \cdot [\text{pRaf} : \text{dpERK}]$$

$$[\dot{\text{P4}}] = -\text{k}_{29} \cdot [\text{Rafi}] \cdot [\text{P4}] + (\text{k}_{\text{b29}} + \text{k}_{30}) \cdot [\text{Rafi} : \text{P4}]$$

$$[\dot{\text{Rafi:P4}}] = \text{k}_{29} \cdot [\text{Rafi}] \cdot [\text{P4}] - (\text{k}_{\text{b29}} + \text{k}_{30}) \cdot [\text{Rafi} : \text{P4}]$$

**Algebraic Equations for conserved species**

$$\text{Raf}_{\text{tot}} = [\text{Raf}] + [\text{Raf:RasGTP}] + [\text{pRaf}] + [\text{pRaf:P1}] + [\text{MEK:pRaf}] + [\text{pMEK:pRaf}] + [\text{Raf:dpERK}] + [\text{Rafi}] + [\text{pRaf:dpERK}] + [\text{Rafi:P4}]$$

$$\text{MEK}_{\text{tot}} = [\text{MEK}] + [\text{MEK:pRaf}] + [\text{pMEK}] + [\text{pMEK:pRaf}] + [\text{dpMEK}] + [\text{dpMEK:P2}] + [\text{pMEK:P2}] + [\text{ERK:dpMEK}] + [\text{pERK:dpMEK}]$$

$$\text{ERK}_{\text{tot}} = [\text{ERK}] + [\text{ERK:dpMEK}] + [\text{pERK}] + [\text{pERK:dpMEK}] + [\text{dpERK}] + [\text{pERK:P3}] + [\text{dpERK:P3}] + [\text{Raf:dpERK}] + [\text{pRaf:dpERK}]$$

$$\text{P1}_{\text{tot}} = [\text{P1}] + [\text{pRaf:P1}]$$

$$\text{P2}_{\text{tot}} = [\text{P2}] + [\text{dpMEK:P2}] + [\text{pMEK:P2}]$$

$$\text{P3}_{\text{tot}} = [\text{P3}] + [\text{pERK:P3}] + [\text{dpERK:P3}]$$

$$\text{P4}_{\text{tot}} = [\text{P4}] + [\text{Rafi:P4}]$$

DOI: https://doi.org/10.7554/eLife.33617.030

**Appendix 1—table 3.** Parameters, variables and equations of the Tgfβ model.

| Parameter | Label | Value | |
|---|---|---|---|
| Phosphorylation rate of Smad2 | $k_{phos}$ | $4.0 \cdot 10^{-4}$ | $nM^{-1}s^{-1}$ |
| Dephosphorylation rate of Smad2 | $k_{dephos}$ | $6.6 \cdot 10^{-3}$ | $nM^{-1}s^{-1}$ |
| Nuclear import rate of Smad2 | $k_{in2}$ | $2.6 \cdot 10^{-3}$ | $s^{-1}$ |
| Nuclear export rate of Smad2 | $k_{ex2}$ | $5.6 \cdot 10^{-3}$ | $s^{-1}$ |
| Nuclear import rate of Smad4 | $k_{in4}$ | $2.6 \cdot 10^{-3}$ | $s^{-1}$ |
| Nuclear export rate of Smad4 | $k_{ex4}$ | $2.6 \cdot 10^{-3}$ | $s^{-1}$ |
| Smad complex import factor | CIF | 5.7 | |
| Forward rate for Smad complex binding | $k_{on}$ | $1.8 \cdot 10^{-3}$ | $nM^{-1}s^{-1}$ |
| Reverse rate for Smad complex binding | $k_{off}$ | $1.6 \cdot 10^{-2}$ | $s^{-1}$ |
| Cytoplasmic to nuclear volume ratio | a | 2.3 | |
| Total Smad2 (initialized to cytoplasm) | $S2_{tot}$ | 73.0 | nM |
| Total Smad4 (initialized to cytoplasm) | $S4_{tot}$ | 73.0 | nM |
| Total phosphatase in nucleus | PPase | 1 | nM |
| Total Receptors | $R_{tot}$ | 1 | nM |
| **Variable** | **Label** | | |
| Cytoplasmic Smad2 | S2c | | |
| Cytoplasmic phosphorylated Smad2 | pS2c | | |
| Cytoplasmic Smad4 | S4c | | |
| Cytoplasmic Smad2:Smad4 complex | S24c | | |
| Cytoplasmic Smad2:Smad2 complex | S22c | | |
| Nuclear Smad2 | S2n | | |
| Nuclear phosphorylated Smad2 | pS2n | | |
| Nuclear Smad4 | S4n | | |
| Nuclear Smad2:Smad4 complex | S24n | | |
| Nuclear Smad2:Smad2 complex | S22n | | |

Differential Equations

$$[\dot{S2c}] = -k_{phos} \cdot u(Tgf\beta) \cdot [S2c] - k_{in2} \cdot [S2c] + k_{ex2} \cdot [S2n]$$

$$[\dot{pS2c}] = \quad k_{phos} \cdot u(Tfg\beta) \cdot [S2c] - k_{in2} \cdot [pS2c] - k_{on} \cdot [pS2c] \cdot ([S4c] + 2 \cdot [pS2c]) + k_{off}$$
$$\cdot ([S24c] + 2 \cdot [S22c]) + k_{ex2} \cdot [pS2n]$$

$$[\dot{S4c}] = -k_{in4} \cdot [S4c] - k_{on} \cdot [pS2c] \cdot [S4c] + k_{off} \cdot [S24c] + k_{ex4} \cdot [S4n]$$

$$[\dot{S24c}] = k_{on}[pS2c] \cdot [S4c] - k_{off} \cdot [S24c] - k_{in2} \cdot CIF \cdot [S24c]$$

$$[\dot{S22c}] = k_{on} \cdot [pS2c]2 - k_{off} \cdot [S22c] - k_{in2} \cdot CIF \cdot [S22c]$$

$$[\dot{S2n}] = a \cdot k_{in2} \cdot [S2c] - a \cdot k_{ex2} \cdot [S2n] + k_{dephos} \cdot PPase \cdot [pS2n]$$

$$[\dot{pS2n}] = \quad a \cdot k_{in2} \cdot [pS2c] - a \cdot k_{ex2} \cdot [pS2n] - k_{dephos} \cdot PPase \cdot [pS2n] - k_{on} \cdot [pS2n]$$
$$\cdot ([S4n] + 2 \cdot [pS2n]) + k_{off} \cdot ([S24n] + 2 \cdot [S22n])$$

*Appendix 1—table 3 continued on next page*

*Appendix 1—table 3 continued*

| Parameter | Label | Value |
|---|---|---|
| $[\dot{S4n}] = a \cdot k_{in4} \cdot [S4c] - a \cdot k_{ex4} \cdot [S4n] - k_{on} \cdot [pS2n] \cdot [S4n] + k_{off} \cdot [S24n]$ | | |
| $[\dot{S24n}] = a \cdot k_{in2} \cdot CIF \cdot [S24c] + k_{on} \cdot [pS2n] \cdot [S4n] - k_{off} \cdot [S24n]$ | | |
| $[\dot{S22n}] = a \cdot k_{in2} \cdot CIF \cdot [S22c] + k_{on} \cdot [pS2n]2 - k_{off} \cdot [S22n]$ | | |
| **Algebraic Equations for conserved species** | | |
| $S2_{tot} = [S2c] + [pS2c] + [S24c] + 2 \cdot [S22c] + (2 \cdot [S22n] + [S24n] + [pS2n] + [S2n])$ | | |
| $S4_{tot} = [S4c] + [S24c] + \frac{1}{a}([S24n] + [S4n])$ | | |

DOI: https://doi.org/10.7554/eLife.33617.031

**Appendix 1—table 4.** Examples of biological systems where the Wnt, ERK, and Tgfβ pathways have been shown to produce graded response in single-cell level.

| Pathway | Systems where graded response has been observed | References |
|---|---|---|
| *Live imaging of single cells* | | |
| Tgfβ pathway | Mouse myoblasts | *Frick et al., 2017; Warmflash et al. (2012)* |
| | Human epidermal keratinocytes | *Nicolás et al. (2004); Warmflash et al. (2012) ; Schmierer et al. (2008); Vizán et al. (2013)* |
| | Human cervical epithelial cells | *Nicolás et al. (2004)* |
| | Human breast epithelial cells | *Strasen et al. (2018)* |
| Canonical Wnt pathway | Human embryonic kidney cells | *Kafri et al. (2016)* (this is the only published live single-cell imaging study in the Wnt pathway so far) |
| ERK pathway | Mouse fibroblasts | *Toettcher et al. (2013)* |
| | Mouse embryonic fibroblasts | *Mackeigan et al. (2005)* |
| | Human non-small cell lung carcinoma | *Cheong et al., 2011* |
| | Human mammary gland cells | *Selimkhanov et al. (2014); Perrett et al., 2013Perrett et al., 2013* |
| | Human cervical epithelial cells | *Voliotis et al. (2014); Whitehurst et al. (2004); Perrett et al., 2013Perrett et al., 2013* |
| | Human foreskin fibroblasts | *Whitehurst et al. (2004)* |
| *Immunofluorescence and FACS studies* | | |
| Tgfβ pathway | *Xenopus* embryo | *Schohl and Fagotto (2002)* |
| | Mouse testes | *Itman et al. (2009)* |
| | Zebrafish embryo | *Dubrulle et al. (2015)* |
| Canonical Wnt pathway | *Xenopus* embryo | *Schneider et al. (1996); Fagotto and Gumbiner (1994); Schohl and Fagotto (2002)* |
| | Mouse embyo | *Aulehla et al., 2008* |
| | Planaria | *Sureda-Gómez et al., 2016* |
| | Sea anemone embryo | *Wikramanayake et al., 2003* |

*Appendix 1—table 4 continued*

| Pathway | Systems where graded response has been observed | References |
|---|---|---|
| ERK pathway | Chick embryo | *Delfini et al. (2005)* |
| | *Xenopus* embryo | *Schohl and Fagotto (2002)* |
| | Human T lymphocyte cells | *Lin et al. (2009)* |
| | Rat adrenal gland cells | *Santos et al., 2007* |

DOI: https://doi.org/10.7554/eLife.33617.032

