## [Decision Letter]

Thank you for submitting your work entitled "Signaling pathways as linear transmitters" for consideration by *eLife*. Your article has been reviewed by three peer reviewers, including Wenying Shou as the Reviewing Editor and Reviewer #1, and the evaluation has been overseen by a Senior Editor. The following individuals involved in review of your submission have agreed to reveal their identity: Steven S Andrews (Reviewer #2).

Our decision has been reached after consultation between the reviewers. We all found your work interesting, but we all had problems with the precise meaning of "physiological ranges".

*Reviewer #1:*

Nunns and Goentoro examined the conserved cores of three signaling pathways (Wnt, ERK, and TGFb). Using mathematical modeling, they found that despite differences in pathway architecture, the three pathways all behaved like linear signal transmitters within physiological ranges. Using experiments to test two pathways (Wnt and ERK), they indeed found linear input-output relationship.

I like how they used dimensionless analysis to gain insights on mechanisms of linearity. My major comments are:

1) They need to describe more carefully the "physiology range": what do they mean by physiological? Moreover, I assume that every parameter has its own range, and dimensionless numbers comprising many parameters probably have a quite large range. What are their ranges? Does most of the range fall within "physiological range"?

2) On a related note, for ERK which is known to sometimes exhibit switch-like responses, how would those parameters and inputs fare in your analytics? Would they have told you that the response will be non-linear?

3) Is there any way to make the wiring diagrams more intuitive and the important points more obvious? Perhaps color coding those parameters in α or γ? More details in legend? For example, why α represents β-catenin degradation by DC is not clear: k11 – which I assume means degradation of phosphor β catenin – is not in α. Why not? The information might already be in supplementary data, but intuition about an expression should be in the main text if you want to appeal to a broad audience.

*Reviewer #2:*

The authors used modeling and then experiment to show linear input-output relations in several model signaling pathways, which were the canonical Wnt, ERK, and Tgf-β pathways. The modeling started with established ODE-based models and reduced them to simplified analytical expressions that related output to input; these expressions did not show linear relationships in general but did in physiologically relevant parameter regimes. The experiments showed linearity in the Wnt and ERK pathways, and that this linearity was reduced through system perturbations.

This is very good work on an important problem. The methods and analysis were appropriate, and the paper is well written.

1) It should be clarified whether the authors are referring to linear signal transmission relative to the signaling pathway ligand concentrations, or to the fraction of bound receptors. This issue arises because the authors show linearity relative to the ligand concentrations in the model analysis portion of this work, but then linearity relative to the fraction of bound receptors in the experimental work on the Wnt pathway. More specifically, it appears that the use of Wnt as the x-axis in Figure 2D contradicts statements in subsection “Linearity in the Wnt and ERK pathways was observed experimentally”, which say that "linearity does not extend upstream to Wnt dose".

2) My experience with cell signaling pathway ODE models is that they capture the known biological interactions reasonably correctly, they agree well with a specific set of test data, and they get the signaling dynamics qualitatively correct, but they are rarely accurate enough for quantitative predictions on untested problems. However, this work uses three models for quantitative predictions. On the one hand, I applaud this approach because it fulfills a major objective of modeling research. On the other hand, I feel that more work is required to convince the reader that the models are in fact accurate enough to warrant the conclusions that are reached from them. This includes the models having sufficient accuracy over both the parameter and time ranges that are considered. For example, if the grey lines in Figure 3 were model calculations (without any additional fitting), I would have greater trust in the models.

3) A persistent concern that I had is that most dose-response functions for cell signaling can be modeled well by Hill functions, and Hill functions are linear relationships in the small dose regime. Is the linearity observed here simply an observation of Hill functions in the small dose regime? If so, then all of these results are reasonably trivial. As part of addressing this issue, it would help if the authors could give the physiologically typical concentration ranges for Wnt, EGF, and Tgf-β.

*Reviewer #3:*

This manuscript aims to test whether signaling pathways have linear input-output response characteristics. A simplification of existing mathematical models indicates this being the case for the canonical Wnt, ERK and TGF-β pathways, despite striking differences in core network architectures among these pathways. It is interesting to see how different complex pathways can be approximated by linear systems. The manuscript concludes by experiments (Western Blots) testing the theoretical results, as well as the breakdown of linearity as some key features of the pathways are altered.

In summary the major point here seems to be that in many of these diverse, complicated, pathways, a linear dose-response can be observed as an approximation, and this might be borne out in experimental conditions as well. In fact, not only can the linear dose response be observed, it appears rather robust to perturbations until you exceed some critical value. While these results are potentially interesting, there are major weaknesses that should be addressed before the manuscript could be considered for publication in *eLife*.

1) Besides the three pathways chosen, there are other pathways that are similarly well studied, both experimentally as well as mathematically. The NF-kappaB pathway is a good example. Why was the NF-kappaB pathway or any other pathway not considered? You could make a statement that proving the generality of this finding will require investigating additional pathways in the future.

2) A linear input-output relationship due to negative feedback has already been described for a MAP Kinase pathway, see PMID:19079053. How are the current results novel compared to these earlier findings (besides studying a human MAPK pathway)?

3) The claims of linearity in this manuscript are mainly based on visual examination. However, there are rigorous ways to measure linearity, based on the L1-norm, as described in PMID:19279212 and PMID:23385595, which should be cited. It would be necessary to apply the L1-norm throughout all figures of the manuscript to make computational and experimental claims of linearity quantitative.

4) For deriving formula [4], apha/u>>1 is necessary. However, based on the SI, α/u=11, and it is actually smaller for some of the range plotted in Figure 2. Then the linearity arises from 'α / u >> 1 + γ'. One could/should explore when α = (1+γ)/u * S where S is a scaling factor set to say, 0.9, 1, 1.1, 2, 10, and 100. At 10 and 100, one might expect the condition to hold, but at S around 1 the simplification probably fails. Do we then see lack of linearity? One can then re-arrange the terms for each of the other terms (u or γ) and explore similarly. The same argument can be made for the Erk and Tgf[β] pathways.

4) How robust is linearity? To answer this question, two modifications are needed: (i) extend the range of u for each plot in Figure 2, showing the nonlinearity of the curves, then indicate the linear range on such nonlinear curves, e.g. with a different color; (ii) Change a parameter and plot a family of curves, indicating the linear range (determined based on the L1-norm) on each individual curve. This can be done for a couple of parameters, but it is especially important for the parameters mediating interactions that cause linearity to break down (described in subsection “Linearity in the Wnt and ERK pathways is modulated by perturbation to parameters”).

5) If linear responses indeed help cellular signal processing as a result of convergent evolution then linearity should probably occur in single cells that process inputs. Unfortunately, the Western Blots in Figure 3 only test linearity at the population average level. The linearity of the average does not imply linearity in single cells. So, are single cell input-output response characteristics linear? This should be checked at least for one signaling pathway, possibly by new experiments or otherwise using data by others, see for example PMID:25504722. In fact, recent papers indicate that some pathways' responses are dynamic/oscillatory, noisy or bimodal at the single cell level. All of this should be addressed/discussed. Where is then the linearity?

6) What defines a "physiological range"? As noted, ultrasensitivity has been described for the ERK pathway's input response. Ultrasensitivity seems contradictory to linearity. Is then ultrasensitivity not physiological? Moreover, what is physiological in a given tissue or organism (from yeast to frog eggs to worms to humans) might not be physiological in another one. All of this should be discussed and clarified.

7) Considering the importance of fold-change detection in biology, it would be important to assess the number of decades over which linearity holds for each pathway. In fact, any nonlinear function can be approximated by a linear relationship except near extremum points (this is the essence of the Taylor approximation). Are these observations more than Taylor expansion? Maybe we can tell based on the decades over which linearity holds.

8) The data in Figure 3 should be expanded into the domains of nonlinearity/saturation and then the range of linearity should be indicated on top of such curves (linearity measured using the L1-norm). This should be done for all panels. In fact, without showing the full curve (including the nonlinear parts), panel 3C does not indicate that linearity is lost or that its range shrinks.

9) What ranges on the theoretical plots do the experimental results correspond to? Theory and experiment should be better connected.

10) Experimental verification is not trivial in these systems. First, the "quantitative' Western blots are not truly quantitative, despite ensuring that fluorescent antibodies are within linear range, etc. Western blots, by their very nature, are notoriously noisy, and therefore require a lot of "normalization" which might artifactually introduce the appearance of linearity. If, for example, the denominator is very large for normalization, the signal can appear more linear. Say, if you compare y=x and y=x^2, but then divide by a large factor K such that within the range of 'x' being explored K>>y, then both equations will be nearly 0 and will likely show a "linear" response (also true according to the Taylor expansion near 0).

11) The variability in the Western blots is borne out in Figure 3A where the various 'Wnt' doses have very variable pLRP5/6 response (input). Likewise, in Figure 3B, for EGF. These few points do not necessarily demonstrate linearity other than the fact that the linear approximation can be applied to just about any relationship. In addition, comparing Figure 3A with Figure 3D where linearity breaks down with a Raf mutation preventing feedback, it still appears linear until 3ng/mL, a dose that is not reported for the wild-type (Figure 3A). In fact, examining both plots between 0 to 2ng/mL, one could conclude instead that both systems are 'linear'.

12) Same with the Wnt pathway. Here it is already recognized that linearity was not lost, but this is attributed to side effects of the drug used – something that could be tested perhaps with a different drug or a mutation similar to the Erk pathway analysis.

---

## [Author Response]

We have now addressed the editors’ and reviewers’ comments, and believe that the manuscript has been made stronger and more precise by the process. We therefore would love to have the manuscript reconsidered, as we believe *eLife* would provide a good home for reaching the suitable audience for the work. Specifically, we have addressed the primary concern regarding the precise meaning of “physiological range”. We agree with the editor and reviewers that this gives a misleading impression that we have information about the parameter ranges across all biological contexts. We have now revised the misleading phrase into “measured parameters in (the name of the specific systems)”, “physiologically relevant parameter ranges”, or “some physiological contexts”.

Reviewer #1:

Nunns and Goentoro examined the conserved cores of three signaling pathways (Wnt, ERK, and TGFb). Using mathematical modeling, they found that despite differences in pathway architecture, the three pathways all behaved like linear signal transmitters within physiological ranges. Using experiments to test two pathways (Wnt and ERK), they indeed found linear input-output relationship.I like how they used dimensionless analysis to gain insights on mechanisms of linearity. My major comments are:1) They need to describe more carefully the "physiology range": what do they mean by physiological? Moreover, I assume that every parameter has its own range, and dimensionless numbers comprising many parameters probably have a quite large range. What are their ranges? Does most of the range fall within "physiological range"?

We see now that the phrasing “physiological range” gives a misleading impression that we have information about the physiological range of the parameters across all biological contexts. To avoid this confusion, we now described it more clearly in the manuscript that we considered how the analytical solutions behaved in a physiologically relevant parameter regime – *i.e.*, the values of parameters that have been measured or estimated in specific biological contexts. All changes are in marked in red. As linearity emerges with these measured parameters, it argues for the physiological relevance of the predicted linearity. Additionally, we experimentally observed linearity in the two human cell lines.

While we have the parameter values that have been measured in some biological contexts, we do not have the full range of the parameter groups across physiological contexts. However, as also requested by reviewer 3 (Point 5), we now include robustness analysis that shows that linearity occurs through a considerable parameter range in the models, although not limitless. It therefore further argues the significance that the measured parameters fall within the range where linearity prevails.

2) On a related note, for ERK which is known to sometimes exhibit switch-like responses, how would those parameters and inputs fare in your analytics? Would they have told you that the response will be non-linear?

Switch-like response in the ERK model could be achieved by decreasing β or increasing _𝛼𝛼_, both amount to weakening the negative feedback. The analysis would tell us that the response is nonlinear (see Figure 2—figure supplement 6C), *i.e.,* L-1 norm > 0.1. Consistently, we observed experimentally that mutating Raf that mediates the negative feedback converts linear to nonlinear response (Figure 3D).

3) Is there any way to make the wiring diagrams more intuitive and the important points more obvious? Perhaps color coding those parameters in α or γ? More details in legend? For example, why α represents β-catenin degradation by DC is not clear: k11 – which I assume means degradation of phosphor β catenin – is not in α. Why not? The information might already be in supplementary data, but intuition about an expression should be in the main text if you want to appeal to a broad audience.

We thank the reviewer for this suggestion. We have revised Figure 2. We added in the main text intuitive description of the parameter grouping, with references to the shading. We also revised the legend to explain why certain parameters are not in the parameter groups (*e.g.*, k11 describes an irreversible degradation of phosphorylated β-catenin, and therefore does not affect the level of unphosphorylated _β_-catenin we are solving for).

Reviewer #2:

The authors used modeling and then experiment to show linear input-output relations in several model signaling pathways, which were the canonical Wnt, ERK, and Tgf-β pathways. The modeling started with established ODE-based models and reduced them to simplified analytical expressions that related output to input; these expressions did not show linear relationships in general but did in physiologically relevant parameter regimes. The experiments showed linearity in the Wnt and ERK pathways, and that this linearity was reduced through system perturbations.This is very good work on an important problem. The methods and analysis were appropriate, and the paper is well written.1) It should be clarified whether the authors are referring to linear signal transmission relative to the signaling pathway ligand concentrations, or to the fraction of bound receptors. This issue arises because the authors show linearity relative to the ligand concentrations in the model analysis portion of this work, but then linearity relative to the fraction of bound receptors in the experimental work on the Wnt pathway. More specifically, it appears that the use of Wnt as the x-axis in Figure 2D contradicts statements in subsection “Linearity in the Wnt and ERK pathways was observed experimentally”, which say that "linearity does not extend upstream to Wnt dose".

We agree this could be confusing since the form of the input function u(ligand) is different across the models, and not necessarily equal to ligand concentration. In the Wnt pathway, u(ligand) describes the Wnt-dependent action of Dvl in promoting dissociation of the destruction complex. In the Tgfβ pathway, u(ligand) is the fraction of active receptors. In the ERK pathway, u(ligand) is the concentration of EGF-activated Ras-GTP. To clarify this, we have now made u(ligand) more obvious in two ways:

A) We include u(ligand) function in the x-axis of Figure 2D-F and add brief intuitive description of each u(ligand), as opposed to having this only in the legend previously.

B) We include in the wiring diagrams in Figure 2A-C illustration of the upstream processes that constitute the function u(ligand), to further emphasize that they are not simply ligand concentration.

2) My experience with cell signaling pathway ODE models is that they capture the known biological interactions reasonably correctly, they agree well with a specific set of test data, and they get the signaling dynamics qualitatively correct, but they are rarely accurate enough for quantitative predictions on untested problems. However, this work uses three models for quantitative predictions. On the one hand, I applaud this approach because it fulfills a major objective of modeling research. On the other hand, I feel that more work is required to convince the reader that the models are in fact accurate enough to warrant the conclusions that are reached from them. This includes the models having sufficient accuracy over both the parameter and time ranges that are considered. For example, if the grey lines in Figure 3 were model calculations (without any additional fitting), I would have greater trust in the models.

We agree that we need to clarify that we are not simply observing linearity in a small dose regime. We now explain more explicitly in the text and include Figure 2—figure supplement 2 to emphasize that the predicted linearity extends the entire or almost the entire dynamic rangeof the systems.

3) A persistent concern that I had is that most dose-response functions for cell signaling can be modeled well by Hill functions, and Hill functions are linear relationships in the small dose regime. Is the linearity observed here simply an observation of Hill functions in the small dose regime? If so, then all of these results are reasonably trivial. As part of addressing this issue, it would help if the authors could give the physiologically typical concentration ranges for Wnt, EGF, and Tgf-β.

Based on our own experience, we hold the same view that when working across biological contexts, we use the models of signaling pathways for exploring qualitative, systemslevel behaviors. With regard to this, the three models have track record of success:

- The ERK model predicts ultrasensitive behavior (Huang and Ferrell, 1996).

- The Wnt model enables deducing the differential roles of the two scaffolds in the pathway (Lee et al., 2003), and the prediction of fold-change robustness (Goentoro and Kirschner, 2009).

- The Tgfβ model reveals the roles of Smad nucleocytoplasmic shuttling in transmitting signaling dynamics (Schmierer et al., 2008).

We now include this information in the Introduction, to provide the readers with more evidence of the predictive power of the models.

Although the models capture systems-level behaviors, they do not necessarily reproduce fine-tuned aspects, e.g., response magnitude or the time to reach steady state, which may quantitatively vary across specific contexts. The prediction in this present study falls within this type of qualitative prediction: the models predict linear behavior, but do not necessarily capture fine-tuned aspects such as the response gain (i.e., slope of the input-output relationship).

Consistently, our aim in Figure 3 was to assess whether the measured input-output relationship was linear or not. We realize now looking at the previous version of Figure 3, that it may have come across we were making quantitative predictions, e.g., fitting the concentrations of EGF. We would like to clarify that we did not fit protein levels quantitatively, and nor does our conclusion depend on such a quantitative fit. The grey lines in Figure 3 are least squares regression lines. We have now made this more obvious in Figure 3 and the legend.

For Figure 3D specifically, we did use the model to fit the data, in the following way. Since a linear fit does not explain the data well, we tested whether the data are better captured by a nonlinear fit. As a nonlinear fit, rather than testing ad-hoc nonlinear functions, we used the full ERK model. We first fitted the gain of the model to the data (i.e., the y-range), and afterward, the only parameter varied was the strength of the negative feedback (k_25_) – since we mutated the negative feedback in this experiment. Using the Akaike Information Criterion (AICc), we verified that a nonlinear fit using the ERK model outperforms a least squares linear fit.

Reviewer #3:

This manuscript aims to test whether signaling pathways have linear input-output response characteristics. A simplification of existing mathematical models indicates this being the case for the canonical Wnt, ERK and TGF-β pathways, despite striking differences in core network architectures among these pathways. It is interesting to see how different complex pathways can be approximated by linear systems. The manuscript concludes by experiments (Western Blots) testing the theoretical results, as well as the breakdown of linearity as some key features of the pathways are altered.In summary the major point here seems to be that in many of these diverse, complicated, pathways, a linear dose-response can be observed as an approximation, and this might be borne out in experimental conditions as well. In fact, not only can the linear dose response be observed, it appears rather robust to perturbations until you exceed some critical value. While these results are potentially interesting, there are major weaknesses that should be addressed before the manuscript could be considered for publication in eLife.1) Besides the three pathways chosen, there are other pathways that are similarly well studied, both experimentally as well as mathematically. The NF-kappaB pathway is a good example. Why was the NF-kappaB pathway or any other pathway not considered? You could make a statement that proving the generality of this finding will require investigating additional pathways in the future.

We did not include the NF-κB pathway because it is analytically intractable in our hands. However, we agree with reviewer 3 that discussion of the NF-κB pathway could help motivate testing linearity in other signaling pathways, even when analytics is not possible. We numerically solved the NF-κB model using parameters that have been measured or fitted in cells. We found that, despite having strong nonlinearities e.g., sequestration, the NF-κB model shows a linear input-output relationship over a physiologically relevant input range. This further strengthens the finding of convergent linearity across multiple complex signaling pathways. We now include this finding in Discussion section and Figure 3—figure supplement 7.

Additional paragraph in Discussion section:

“It would be interesting to further probe the generality of linear signal transmission. … Besides the systems analyzed here, NF-_κB_ is another signaling pathway that has been modeled rigorously (5, 50-51). Numerical simulations of a well-established NF-_κB_ model (50) over the range of nuclear NF-_κB_ translocation observed in human epithelial cells (51) reveal that the peak of the nuclear NF-κB pulse correlates linearly with ligand concentration (Figure 2—figure supplement 7).”

2) A linear input-output relationship due to negative feedback has already been described for a MAP Kinase pathway, see PMID:19079053. How are the current results novel compared to these earlier findings (besides studying a human MAPK pathway)?

To begin with, these are two distinct MAPK pathways: Fus3(MAPK) pathway in fungal yeast is coupled to G protein-coupled receptor (GPCR), whereas we are considering ERK(MAPK) pathway in mammals coupled to receptor tyrosine kinases. Even if we simply consider negative feedback broadly, the present results are novel in two ways:

1) The negative feedbacks between the fungal and animal MAPK pathways, as currently known, are convergent. Yu et al. 2008 (Richard et al., 2008) identified negative feedback by Fus3 acting on Sst2, a yeast GAP protein. It is known that ERK feedbacks on GRK, a kinase upstream from GAP, but no feedbacks have been identified on mammalian GAPs. Even the newly identified feedback actions of Fus3, on Ste5 and Ste18 (Choudhury, Baradaran-Mashinchi and Torres, 2018), are not conserved in mammals. Finally, with regard to the specific feedback we are considering here, where ERK hyper-phosphorylates Raf, there is no known homolog of Raf in yeasts.

2) The linearity we identified in the EGF/ERK model here not only requires negative feedback, but also ultrasensitivity in the Raf-MEK-ERK cascade.

These differences further strengthen the finding of convergent linearity across independently evolving signaling pathways, as well as homologous pathways that diverged 1.5 billion years ago. We now include this in Discussion section:

“Finally, linearity extends beyond metazoan signaling pathways. In the yeast pheromone sensing pathway, a homolog of the ERK cascade, transcriptional output correlates linearly with receptor occupancy (Richard et al., 2008). The linearity is mediated by negative feedback by Fus3 acting on Sst2, a feedback that is not conserved in the mammalian ERK system. These further argue for linear signal transmission as a convergent property across independently evolving signaling pathways, as well as between conserved pathways that diverged 1.5 billion years ago.”

3) The claims of linearity in this manuscript are mainly based on visual examination. However, there are rigorous ways to measure linearity, based on the L1-norm, as described in PMID:19279212 and PMID:23385595, which should be cited. It would be necessary to apply the L1-norm throughout all figures of the manuscript to make computational and experimental claims of linearity quantitative.

We apologize that some information was missing from Figure 3 that gives the impression that we assessed the linearity only visually. We have now noted more clearly in Figure 3 that we applied for each dataset least squares regression analysis (L2-norm). As there is complementarity between L1- and L2-norm, we also follow reviewer 3’s suggestion and added L1-norm analysis for all linearity assessment in the manuscript. The conclusions hold with both analyses.

(We now address Point 7 and 8 first, since these points are relevant to many other questions.)

7) Considering the importance of fold-change detection in biology, it would be important to assess the number of decades over which linearity holds for each pathway. In fact, any nonlinear function can be approximated by a linear relationship except near extremum points (this is the essence of the Taylor approximation). Are these observations more than Taylor expansion? Maybe we can tell based on the decades over which linearity holds.

As reviewer 1 also suggests, we see now that the phrase “physiological range” creates a misleading impression that we have information about the entire physiological range of the parameters across tissues and organisms. We have now described more carefully in the manuscript that we considered a physiological relevant parameter regime, i.e., the values of parameters that have been measured in some biological systems, that we specify in the main text. As linearity emerges with these measured parameters, it argues for the physiological relevance of the predicted linearity. Indeed, we observed linear input-output relationship in two human cell lines.

With this revision, it should be clear now that we do not suggest that linearity occurs in all possible physiological ranges, but in some physiological contexts. As also raised by reviewer 1 (Point 2), the ERK pathway produces ultrasensitive response in some biological contexts, as well as graded response in some other contexts. Both responses can be captured by modulating the negative feedback in the ERK model. Finally, we now include a section in the supplement entitled “Toy Model of the ERK pathway” that illustrates how ultrasensitivity can produce linearity, to further clarify their seemingly contradictory relationship.

8) The data in Figure 3 should be expanded into the domains of nonlinearity/saturation and then the range of linearity should be indicated on top of such curves (linearity measured using the L1-norm). This should be done for all panels. In fact, without showing the full curve (including the nonlinear parts), panel 3C does not indicate that linearity is lost or that its range shrinks.

As also suggested by reviewer 2, we now explain more explicitly that the predicted linearity extends over the dynamic range of the model, theoretical or observed (Figure 2—figure supplement 2).

Across these three natural pathways, the dynamic range may not appear as impressive as e.g., some synthetic circuits that can operate across multiple orders of magnitudes. And yet, this is one reason, we think, fold-change detection may be important, as it allows the system to continually rescale its dynamic range to present stimulation level, and maintains sensitivity to respond (as discussed in ref. (Olsman and Goentoro, 2016)).

4) For deriving formula [4], apha/u>>1 is necessary. However, based on the SI, α/u=11, and it is actually smaller for some of the range plotted in Figure 2. Then the linearity arises from 'α / u >> 1 + γ'. One could/should explore when α = (1+γ)/u * S where S is a scaling factor set to say, 0.9, 1, 1.1, 2, 10, and 100. At 10 and 100, one might expect the condition to hold, but at S around 1 the simplification probably fails. Do we then see lack of linearity? One can then re-arrange the terms for each of the other terms (u or γ) and explore similarly. The same argument can be made for the Erk and Tgf[β] pathways.

We thank reviewer 3, we now include this analysis as Figure 2—figure supplement 5.

5) How robust is linearity? To answer this question, two modifications are needed: (i) extend the range of u for each plot in Figure 2, showing the nonlinearity of the curves, then indicate the linear range on such nonlinear curves, e.g. with a different color; (ii) Change a parameter and plot a family of curves, indicating the linear range (determined based on the L1-norm) on each individual curve. This can be done for a couple of parameters, but it is especially important for the parameters mediating interactions that cause linearity to break down (described in subsection “Linearity in the Wnt and ERK pathways is modulated by perturbation to parameters”).

We agree this are a good analysis to include. For the input range, as discussed in Point 8 above, the linearity extends the dynamic range of the models. For parameter variation, we now include this analysis as a supplement figure. In these simulations, we plot input-output relationship over the entire dynamic range of the models in unperturbed system (as discussed in Point 8). Shaded in purple is the region of the response with L1-norm < 0.1. The analysis shows that linearity occurs through a considerable range of parameter values.

6) If linear responses indeed help cellular signal processing as a result of convergent evolution then linearity should probably occur in single cells that process inputs. Unfortunately, the Western Blots in Figure 3 only test linearity at the population average level. The linearity of the average does not imply linearity in single cells. So, are single cell input-output response characteristics linear? This should be checked at least for one signaling pathway, possibly by new experiments or otherwise using data by others, see for example PMID:25504722. In fact, recent papers indicate that some pathways' responses are dynamic/oscillatory, noisy or bimodal at the single cell level. All of this should be addressed/discussed. Where is then the linearity?

We agree with the reviewer that discussing this point is necessary. While quantitative assessment of single-cell dynamics is now increasingly done, double quantitation of input-output in single cells is still technically challenging. It is even more challenging in our study because the inputs and outputs here are not simply protein levels, but include phosphorylated proteins (e.g., pLRP, dpERK). On the advice of the reviewer, we examined existing single-cell data from ref. (Selimkhanov et al., 2014), kindly provided by Roy Wollman, which utilizes a FRET sensor to measure ERK activity. The dataset offers useful insights into the dynamics of single-cell ERK response (as discussed in their paper); however, the dynamic range of the sensor was not large enough to distinguish between dose-response models (<2-fold, compared to 12-fold for our Western Blot data). Another method, quantitative dual-immunofluorescence, requires having at least two good linear antibodies against each target of interest to calibrate the linearity of the antibodies (e.g., what was done in ref. (Cheong et al., 2011)), and unfortunately, which are not available for the targets of interest here.

Despite these technical limitations, we can still address reviewer’s concern in 2 ways. First, we can qualitatively assess existing data from single cells for the plausibility of linearity in some biological contexts. Linear behavior means that single cells responds to ligand in a graded manner. Even though there are reports of oscillatory or bimodal response in signaling pathways, there are also multiple observations where signaling pathways show graded response in single cells. We provide some of these examples for the three pathways in Supplementary file 4.

We include this information in Discussion section:

“It would be interesting to further probe the generality of linear signal transmission. Linear behavior requires that single cells responds to ligand in a graded manner. Although there are reports of oscillatory or bimodality in signaling pathways, there are also multiple observations across biological contexts of single cells responding to ligand in a graded manner (Table S4).”

Second, the ERK pathway, in particular, is sometime observed to show ultrasensitive response. We confirmed that the particular cells used in this study, non-small cell lung carcinoma, have been observed to show graded response at single-cell level (measured in live cells at the level of total ERK, (Cheong et al., 2011)), with no evidence for bimodality. We further confirmed qualitatively the graded response in these cells using immunofluorescence against dpERK (now included as Figure 3—figure supplement 4) (note as described earlier, we cannot quantify this immunofluorescent signal as we cannot calibrate if the antibody is linear in immunofluorescence assay).

9) The data in Figure 3 should be expanded into the domains of nonlinearity/saturation and then the range of linearity should be indicated on top of such curves (linearity measured using the L1-norm). This should be done for all panels. In fact, without showing the full curve (including the nonlinear parts), panel 3C does not indicate that linearity is lost or that its range shrinks.

As also requested by reviewer 2, we have now included more measurements to further confirm the dynamic range of the cells used in the experiments. The revised Figure 3 is included below, where data points beyond saturation are shown in grey, and linearity is assessed in blue data points.

10) What ranges on the theoretical plots do the experimental results correspond to? Theory and experiment should be better connected.

We have now made the theory-experiment connection clearer by making it more explicit in text and Figure 2—figure supplement 2 that the models predict linearity across the dynamic range of the systems (as detailed in Point 8). Corresponding to the model prediction, we measured the entire dynamic range of the cells and observed that linearity governs the entire range until saturation (as detailed in Point 9).

11) Experimental verification is not trivial in these systems. First, the "quantitative' Western blots are not truly quantitative, despite ensuring that fluorescent antibodies are within linear range, etc. Western blots, by their very nature, are notoriously noisy, and therefore require a lot of "normalization" which might artifactually introduce the appearance of linearity. If, for example, the denominator is very large for normalization, the signal can appear more linear. Say, if you compare y=x and y=x^2, but then divide by a large factor K such that within the range of 'x' being explored K>>y, then both equations will be nearly 0 and will likely show a "linear" response (also true according to the Taylor expansion near 0).

To show that the linearity is not a caveat of normalization in the measurements, here are some raw data without loading control normalization, in experiments where the loading control across the samples came out fairly uniform, <10%. (This can sometime happen because we processed samples in our protocol as uniformly as possible). The observed linearity holds even without normalization.

12) The variability in the Western blots is borne out in Figure 3A where the various 'Wnt' doses have very variable pLRP5/6 response (input). Likewise, in Figure 3B, for EGF. These few points do not necessarily demonstrate linearity other than the fact that the linear approximation can be applied to just about any relationship.

The variability of our Western blot measurement is within 10%, arguing for the precision of our measurements (demonstrated in figure below, now included as Figure 3—figure supplement 7).

While we can tightly control technical variability, there are inevitable biologic differences, e.g., each plate of cells may have slightly different cell confluency, cell contact, cell cycle state and therefore slightly different receptivity to ligands. In addition, measurements of ligand concentrations may also have uncertainty. All this may influence response amplitude (or gain) across experiments, which is also a highly fine-tuned parameter in the models. Despite such variability in response gain across experiments, the linear response is highly reproducible across experiments. The following figure shows some measurements from independent experiments. Moreover, where we can measure the internal proxy for ligand activation (phopho-LRP), regardless of uncertainty in externally measured Wnt dose, it is notable that phospho-LRP5/6 remains linearly correlated with β-catenin.

12) (ii) In addition, comparing Figure 3A with Figure 3D where linearity breaks down with a Raf mutation preventing feedback, it still appears linear until 3ng/mL, a dose that is not reported for the wild-type (Figure 3A). In fact, examining both plots between 0 to 2ng/mL, one could conclude instead that both systems are 'linear'.

We agree that the way we currently use the term ‘linearity’ in the text can be confusing, since any function would trivially have a linear regime for some small range. Following reviewer 2’s suggestion, we have now made it more clearly in the revised manuscript that the predicted linearity spans the dynamic range of the models. Consistently, in testing the prediction in the ERK pathway, we analyzed linearity throughout the dynamic range of the cells (from 0-4 ng/mL EGF). We now include in the main text a data point for 3 ng/mL EGF in wt cells as well. Performing L1-norm analysis over 0-4 ng/mL EGF, we find for wt cells, L1-norm = 0.03, and for mutant cells, L1-norm = 0.26. Thus, one may conclude that upon mutation in the negative feedback linearity was lost or reduced, either way confirming that the negative feedback is necessary to maintain linearity.

13) Same with the Wnt pathway. Here it is already recognized that linearity was not lost, but this is attributed to side effects of the drug used – something that could be tested perhaps with a different drug or a mutation similar to the Erk pathway analysis.

Indeed, in the Wnt pathway, we not only observed linearity, but also that it is unexpectedly more strongly maintained in the cells than the model predicts. This is not a side effect of the drug, but rather a feature of the dual, incoherent role of GSK3β in phosphorylating βcatenin for degradation and phosphorylating LRP for the pathway activation (phospho-LRP inhibits β-catenin degradation). Encoding such an incoherent feedforward loop in the model does capture the maintenance of linearity. We now include the model in Figure 2—figure supplement 4 and add this discussion in the main text (Discussion section). Thus, the experiments not only confirm linear signal transmission in the Wnt pathway, but also reveal a hitertho unknown role of the GSK3β incoherent feedforward loop in maintaining the linearity.